# Demystifying Local & Global Fairness Trade-offs in Federated Learning Using Partial Information Decomposition

**Faisal Hamman**
University of Maryland College Park
fhamman@umd.edu

**Sanghamitra Dutta**
University of Maryland College Park
sanghamd@umd.edu

## Abstract

This work presents an information-theoretic perspective to group fairness trade-offs in federated learning (FL) with respect to sensitive attributes, such as gender, race, etc. Existing works often focus on either *global fairness* (overall disparity of the model across all clients) or *local fairness* (disparity of the model at each client), without always considering their trade-offs. There is a lack of understanding regarding the interplay between global and local fairness in FL, particularly under data heterogeneity, and if and when one implies the other. To address this gap, we leverage a body of work in information theory called partial information decomposition (PID), which first identifies three sources of unfairness in FL, namely, *Unique Disparity*, *Redundant Disparity*, and *Masked Disparity*. We demonstrate how these three disparities contribute to global and local fairness using canonical examples. This decomposition helps us derive fundamental limits on the trade-off between global and local fairness, highlighting where they agree or disagree. We introduce the *Accuracy and Global-Local Fairness Optimality Problem (AGLFOP)*, a convex optimization that defines the theoretical limits of accuracy and fairness trade-offs, identifying the best possible performance any FL strategy can attain given a dataset and client distribution. We also present experimental results on synthetic datasets and the ADULT dataset to support our theoretical findings.[1]

## 1 Introduction

*Federated learning* (FL) is a framework where several parties (*clients*) collectively train machine learning models while retaining the confidentiality of their local data (Yang, 2020; Kairouz et al., 2021). With the growing use of FL in various high-stakes applications, such as finance, healthcare, recommendation systems, etc., it is crucial to ensure that these models do not discriminate against any demographic group based on sensitive features such as race, gender, age, nationality, etc. (Smith et al., 2016). While there are several methods to achieve group fairness in the centralized settings (Mehrabi et al., 2021), these methods do not directly apply to a FL setting since each client only has access to their local dataset, and hence, is restricted to only performing local disparity mitigation.

Recent works (Du et al., 2021; Abay et al., 2020; Ezzeldin et al., 2023) focus on finding models that are fair when evaluated on the entire dataset across all clients, a concept known as *global fairness*. E.g., several banks may decide to engage in FL to train a model that will determine loan qualifications without exchanging data among them. A globally fair model does not discriminate against any protected group when evaluated on the entire dataset across all the banks. On the other hand, *local fairness* considers the disparity of the model at each client (when evaluated on a client's local dataset). Local fairness is important as the models are ultimately deployed and used locally (Cui et al., 2021).

Global and local fairness can differ, particularly when the local demographics at a client differ from the global demographic across the entire dataset (data heterogeneity, e.g., a bank with customers primarily from one race). Prior studies have mainly focused on either global or local fairness, without always considering their interplay. Global and local fairness align when data is i.i.d. across clients (Ezzeldin et al., 2023; Cui et al., 2021), but their interplay in other scenarios is not well-understood.

---

[1]Implementation is available at https://github.com/FaisalHamman/FairFL-PID

This work aims to provide a fundamental understanding of group fairness trade-offs in the FL setting. We first formalize the notions of Global and Local Disparity in FL using information theory. Next, we leverage a body of work within information theory called partial information decomposition (PID) to further identify three sources of disparity in FL that contribute to the Global and Local Disparity, namely, *Unique Disparity*, *Redundant Disparity*, and *Masked Disparity*. This information-theoretic decomposition is significant because it helps us derive fundamental limits on the trade-offs between Global and Local Disparity, particularly under data heterogeneity, and provides insights on when they agree and disagree. We introduce the *Accuracy and Global-Local Fairness Optimality Problem* (AGLFOP), a novel convex optimization that rigorously examines the trade-offs between accuracy and both global and local fairness. This framework establishes the theoretical limits of what any FL technique can achieve in terms of accuracy and fairness *given a dataset and client distribution.* This work provides a more nuanced understanding of the interplay between these two fairness notions that can better inform disparity mitigation techniques and their convergence and effectiveness in practice.

Our main contributions can be summarized as follows:

- **Partial information decomposition of Global and Local Disparity:** We first define Global Disparity as the mutual information $\mathrm{I}(Z; \hat{Y})$ where $\hat{Y}$ is a model's prediction and $Z$ is the sensitive attribute (see Definition 2). Then, we show that Local Disparity can in fact be represented as the conditional mutual information $\mathrm{I}(Z; \hat{Y}|S)$ where $S$ denotes the client (see Definition 3). We also demonstrate relationships between these information-theoretic quantifications and well-known fairness metrics such as statistical parity (see Lemma 1). Using an information-theoretic quantification then enables us to further decompose the Global and Local Disparity into three non-negative components: *Unique*, *Redundant*, and *Masked Disparity*. We provide canonical examples to help understand these disparities in the context of FL (see Section 3.1). The significance of our information-theoretic decomposition lies in separating the regions of agreement and disagreement of Local and Global Disparity, demystifying their trade-offs.

- **Fundamental limits on trade-offs between local and global fairness:** We show the limitations of achieving global fairness using local disparity mitigation techniques due to *Redundant Disparity* (see Theorem 1) and the limitations of achieving local fairness even if global fairness is achieved due to Masked Disparity (see Theorem 2). We also identify the necessary and sufficient conditions under which one form of fairness (local or global) implies the other (see Theorem 3 and 4), as well as, discuss other conditions that are sufficient but not necessary.

- **A convex optimization framework for quantifying accuracy-fairness trade-offs:** We present the *Accuracy and Global-Local Fairness Optimality Problem* (AGLFOP) (see Definition 4), a novel convex optimization framework for systematically exploring the trade-offs between accuracy and both global and local fairness metrics. AGLFOP evaluates all potential joint distributions, thereby setting the theoretical boundaries for the best possible performance achievable for a given dataset and client distribution in FL.

- **Experimental demonstration:** We validate our theoretical findings using synthetic and Adult dataset (Dua & Graff, 2017). We study the trade-offs between accuracy and global-local fairness by examining the Pareto frontiers of the AGLFOP. We investigate the PID of disparities in the Adult dataset trained within a FL setting with multiple clients under various data heterogeneity scenarios.

**Related Works.** There are various perspectives to fairness in FL (Shi et al., 2021). One definition is *client-fairness* (Li et al., 2019), which aims to achieve equal performance across all client devices (Wang et al., 2023b). In this work, we are instead interested in group fairness, i.e., fairness with respect to demographic groups based on gender, race, etc. Methods for achieving group fairness in a centralized setting (Agarwal et al., 2018; Hardt et al., 2016; Dwork et al., 2012; Kamishima et al., 2011; Pessach & Shmueli, 2022) may not directly apply in a FL setting since each client only has access to their local dataset. Existing works on group fairness in FL generally aim to develop models that achieve *global fairness*, without much consideration for the *local fairness* at each client (Ezzeldin et al., 2023). For instance, one approach to achieve global fairness in FL poses a constrained optimization problem to find the best model locally, while also ensuring that disparity at a client does not exceed a threshold and then aggregates those models (Chu et al., 2021; Rodríguez-Gálvez et al., 2021; Zhang et al., 2020). Other techniques involve bi-level optimization that aims to find the optimal global model (minimum loss) under the worst-case fairness violation (Papadaki et al., 2022; Hu et al., 2022; Zeng et al., 2021), or re-weighting mechanisms (Abay et al., 2020; Du et al., 2021), both of which often require sharing additional parameters with a server.

Cui et al. (2021) argues for local fairness, as the model will be deployed at the local client level, and propose constrained multi-objective optimization. While accuracy-fairness tradeoffs have been examined in centralized settings (Chen et al., 2018; Dutta et al., 2020a; Kim et al., 2020; Zhao & Gordon, 2022; Liu & Vicente, 2022; Wang et al., 2023a; Kim et al., 2020; Sabato & Yom-Tov, 2020; Menon & Williamson, 2017; Venkatesh et al., 2021), such considerations, along with the relationship between local and global fairness, remain largely unexplored in FL. Our work addresses this gap by examining them through the lens of PID.

Information-theoretic measures have been used to quantify group fairness in the centralized setting in Kamishima et al. (2011); Calmon et al. (2017); Ghassami et al. (2018); Dutta et al. (2020b; 2021); Cho et al. (2020); Baharlouei et al. (2019); Grari et al. (2019); Wang et al. (2021); Galhotra et al. (2022); Alghamdi et al. (2022); Kairouz et al. (2019); Dutta & Hamman (2023). PID is also generating interest in other ML problems (Ehrlich et al., 2022; Tax et al., 2017; Liang et al., 2024; Wollstadt et al., 2023; Mohamadi et al., 2023; Pakman et al., 2021; Venkatesh & Schamberg, 2022). Here, instead of trying to minimize information-theoretic measures as a regularizer, our goal is to quantify the fundamental trade-offs between local and global fairness in FL and develop insights on their interplay to better understand what is information-theoretically possible using any technique.

## 2 PRELIMINARIES

**Notations.** A client is represented as $S \in \{0, 1, \ldots, K-1\}$, where $K$ is the total number of federating clients. A client $S=s$ has a dataset $\mathcal{D}_s = \{(x_i, y_i, z_i)\}_{i=1,\ldots n_s}$, where $x_i$ denotes the input features, $y_i \in \{0, 1\}$ is the true label, and $z_i \in \{0, 1\}$ is the sensitive attribute (assume binary) with 1 indicating the privileged group and 0 indicating the unprivileged group. The term $n_s$ denotes the number of datapoints at client $S=s$. The collective dataset is given by $\mathcal{D} = \cup_{s=0}^{K-1} \mathcal{D}_s$. When denoting a random variable drawn from this dataset, we let $X$ be the input features, $Z$ be the sensitive attribute, and $Y$ be the true label. We also let $\hat{Y}$ represent the predictions of a model $f_\theta(X)$ which is parameterized by $\theta$.

Standard FL aims to minimize the empirical risk: $\min_\theta L(\theta) = \min_\theta \frac{1}{K} \sum_{s=0}^{K-1} \alpha_s L_s(\theta)$, where $L_s(\theta) = \frac{1}{n_s} \sum_{(x,y) \in \mathcal{D}_s} l(f_\theta(x), y)$ is the local objective (or loss) at client $s$, $\alpha_s$ is an importance coefficient (often equal across clients), and $l(\cdot, \cdot)$ denotes a predefined loss function. To minimize the objective $L(\theta)$, a decentralized approach is employed. Each client $S = s$ trains on their private dataset $\mathcal{D}_s$ and provides their trained local model to a centralized server. The server aggregates the parameters of the local models to create a global model $f_\theta(x)$ (Sah & Singh, 2022). E.g., the *FedAvg* algorithm (McMahan et al., 2017) is a popular approach that aggregates the parameters of local models by taking their average, which is then used to update the global model. This process is repeated until the global model achieves a satisfactory *performance* level.

**Background on Partial Information Decomposition.** PID decomposes the mutual information $I(Z; A, B)$ about a random variable Z contained in the tuple $(A, B)$ into four non-negative terms:

$$I(Z; A, B) = \text{Uni}(Z{:}A|B) + \text{Uni}(Z{:}B|A) + \text{Red}(Z{:}A, B) + \text{Syn}(Z{:}A, B) \qquad (1)$$

Here, $\text{Uni}(Z{:}A|B)$ denotes the unique information about $Z$ that is present only in $A$ and not in $B$. E.g., *shopping preferences* ($A$) may provide unique information about *gender* ($Z$) that is not present in *address* ($B$). $\text{Red}(Z{:}A, B)$ denotes the redundant information about $Z$ that is present in both $A$ and $B$. E.g., *zipcode* ($A$) and *county* ($B$) may provide redundant information about *race*.

$\text{Syn}(Z{:}A, B)$ denotes the synergistic information not present in either $A$ or $B$ individually, but present jointly in $(A, B)$, e.g., each individual digit of the *zipcode* may not have information about *race* but together they provide significant information about *race*.

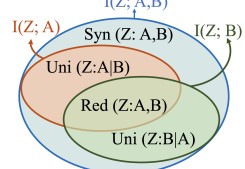

Figure 1: PID of $I(Z; A, B)$.

***Numerical Example.*** Let $Z=(Z_1, Z_2, Z_3)$ with each $Z_i \sim$ i.i.d. Bern($\frac{1}{2}$). Let $A = (Z_1, Z_2, Z_3 \oplus N)$, $B = (Z_2, N)$, $N \sim$ Bern($\frac{1}{2}$) is independent of $Z$. Here, $I(Z; A, B) = 3$ bits. The unique information about $Z$ that is contained only in $A$ and not in $B$ is effectively in $Z_1$, and is given by $\text{Uni}(Z{:}A|B) = I(Z; Z_1) = 1$ bit. The redundant information about $Z$ that is contained in both $A$ and $B$ is effectively in $Z_2$ and is given by $\text{Red}(Z{:}A, B) = I(Z; Z_2) = 1$ bit. Lastly, the synergistic information about $Z$ that is not contained in either $A$ or $B$ alone, but is contained in both of them together is effectively in the tuple $(Z_3 \oplus N, N)$,

and is given by $\mathrm{Syn}(Z{:}A, B){=}\mathrm{I}(Z; (Z_3 \oplus N, N)) = 1$ bit. This accounts for the 3 bits in $\mathrm{I}(Z; A, B)$. Here, we include a popular definition of $\mathrm{Uni}(Z{:}A|B)$ from Bertschinger et al. (2014).

**Definition 1** (Unique Information). *Let $\Delta$ be the set of all joint distributions on $(Z, A, B)$ and $\Delta_p$ be the set of joint distributions with the same marginals on $(Z, A)$ and $(Z, B)$ as the true distribution, i.e., $\Delta_p = \{Q{\in}\Delta : \mathrm{Pr}_Q(Z{=}z, A{=}a){=}\mathrm{Pr}(Z{=}z, A{=}a)$ and $\mathrm{Pr}_Q(Z{=}z, B{=}b) = \mathrm{Pr}(Z{=}z, B{=}b)\}$. Then, $\mathrm{Uni}(Z{:}A|B) = \min_{Q \in \Delta_p} \mathrm{I}_Q(Z; A|B)$, where $\mathrm{I}_Q(Z; A|B)$ is the conditional mutual information when $(Z, A, B)$ have joint distribution $Q$ and $\mathrm{Pr}_Q(\cdot)$ denotes the probability under $Q$.*

Defining any one of the PID terms suffices to obtain the others. $\mathrm{Red}(Z{:}A, B)$ is the sub-volume between $\mathrm{I}(Z; A)$ and $\mathrm{I}(Z; B)$ (see Fig. 1). Hence, $\mathrm{Red}(Z{:}A, B) = \mathrm{I}(Z; A) - \mathrm{Uni}(Z{:}A|B)$ and $\mathrm{Syn}(Z{:}A, B) = \mathrm{I}(Z; A, B) - \mathrm{Uni}(Z{:}A|B) - \mathrm{Uni}(Z{:}B|A) - \mathrm{Red}(Z{:}A, B)$ (from Equation 1).

## 3 MAIN RESULTS

We first formalize the notions of Global and Local Disparity in FL using information theory.

**Definition 2** (Global Disparity). *The Global Disparity of a model $f_\theta$ with respect to $Z$ is defined as $\mathrm{I}(Z; \hat{Y})$, the mutual information between $Z$ and $\hat{Y}$ (where $\hat{Y} = f_\theta(X)$).*

This is related to a widely-used group fairness notion called statistical parity. Existing works (Agarwal et al., 2018) define the Global Statistical Parity as: $\mathrm{Pr}(\hat{Y} = 1|Z = 1) = \mathrm{Pr}(\hat{Y} = 1|Z = 0)$. Global Statistical Parity is satisfied when $Z$ is independent of $\hat{Y}$, which is equivalent to zero mutual information $\mathrm{I}(Z; \hat{Y}) = 0$. To further justify our choice of $\mathrm{I}(Z; \hat{Y})$ as a measure of Global Disparity, we provide a relationship between the absolute statistical parity gap and mutual information when they are non-zero in Lemma 1 (Proof in Appendix B).

**Lemma 1** (Relationship between Global Statistical Parity Gap and $\mathrm{I}(Z; \hat{Y})$). *Let $\mathrm{Pr}(Z{=}0) = \alpha$. The gap $SP_{global} = |\mathrm{Pr}(\hat{Y} = 1|Z = 1) - \mathrm{Pr}(\hat{Y} = 1|Z = 0)|$ is bounded by $\frac{\sqrt{0.5\, \mathrm{I}(Z; \hat{Y})}}{2\alpha(1-\alpha)}$.*

A critical observation that we make in this work is that: *local unfairness can be quantified as $\mathrm{I}(Z; \hat{Y}|S)$, the conditional mutual information between $Z$ and $\hat{Y}$ conditioned on $S$. This is motivated from Ezzeldin et al. (2023) which defines Local Statistical Parity at a client $s$ as:* $\mathrm{Pr}(\hat{Y}{=}1|Z{=}1, S{=}s) = \mathrm{Pr}(\hat{Y}{=}1|Z{=}0, S{=}s)$.

**Definition 3** (Local Disparity). *The Local Disparity is the conditional mutual information $\mathrm{I}(Z; \hat{Y}|S)$.*

**Lemma 2.** $\mathrm{I}(Z; \hat{Y}|S){=}0$ *if and only if* $\mathrm{Pr}(\hat{Y}{=}1|Z{=}1, S{=}s){=}\mathrm{Pr}(\hat{Y}{=}1|Z{=}0, S{=}s)$ *at all clients $s$.*

The proof (see Appendix B) uses the fact that $\mathrm{I}(Z; \hat{Y}|S) = \sum_{s=0}^{K-1} \mathrm{Pr}(S{=}s)\mathrm{I}(Z; \hat{Y}|S = s)$ where $\mathrm{I}(Z; \hat{Y}|S = s)$ is the Local Disparity at client $s$, and $\mathrm{Pr}(S{=}s) = n_s/n$, the proportion of data points at client $s$. Similar to Lemma 1, we can also get a relationship between $SP_s$ and $\mathrm{I}(Z; \hat{Y}|S = s)$ when they are non-zero (see Corollary 1 in Appendix B). We can also define other fairness metrics similarly. For instance, *Global Equalized Odds* can be formulated in terms of the conditional mutual information, denoted as $\mathrm{I}(Z; \hat{Y}|Y)$ and *Local Equalized Odds* as $\mathrm{I}(Z; \hat{Y}|Y, S)$.

### 3.1 PARTIAL INFORMATION DECOMPOSITION OF GLOBAL AND LOCAL DISPARITY

We provide a decomposition of Global and Local Disparity into three sources of unfairness: Unique, Redundant, and Masked Disparity, and provide examples to illustrate and better understand these disparities in the context of FL.

**Proposition 1.** *The Global and Local Disparity in FL can be decomposed into non-negative terms:*

$$\mathrm{I}(Z; \hat{Y}) = \mathrm{Uni}(Z{:}\hat{Y}|S) + \mathrm{Red}(Z{:}\hat{Y}, S). \tag{2}$$

$$\mathrm{I}(Z; \hat{Y}|S) = \mathrm{Uni}(Z{:}\hat{Y}|S) + \mathrm{Syn}(Z{:}\hat{Y}, S). \tag{3}$$

We refer to Fig. 2 for an illustration of this result. Equation 2 follows from the relationship between different PID terms while Equation 3 requires the chain rule of mutual information (Cover, 1999). For completeness, we show the non-negativity of PID terms in Appendix C.

The term $\mathrm{Uni}(Z{:}\hat{Y}|S)$ quantifies the unique information the sensitive attribute $Z$ provides about the model prediction $\hat{Y}$ that is not provided by client label $S$. We refer to this as the ***Unique Disparity***. The Unique Disparity contributes to both Local and Global Disparity, highlighting the region where they agree. $\mathrm{Red}(Z{:}\hat{Y},S)$ quantifies the information about sensitive attribute $Z$ that is common between prediction $\hat{Y}$ and client $S$. We call this the ***Redundant Disparity***. The Unique and Redundant Disparities together make up the Global Disparity $\mathrm{I}(Z;\hat{Y})$. $\mathrm{Syn}(Z{:}\hat{Y},S)$ represents the synergistic information about sensitive attribute $Z$ that is *not* present in either $\hat{Y}$ or $S$ individually, but is present jointly in $(\hat{Y},S)$. We refer to this as the ***Masked Disparity***, as it is only observed when $\hat{Y}$ and $S$ are considered together. Redundant and Masked Disparities cause disagreement between global and local fairness.

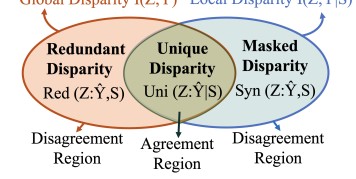

Figure 2: Venn diagram of PID for Global & Local Disp. with agreement and disagreement regions.

**Canonical Examples.** We now examine a loan approval scenario featuring binary-sensitive attributes across two clients, i.e., $\hat{Y}, Z, S \in \{0,1\}$. Here, $\mathrm{I}(Z;\hat{Y},S)=H(Z)-H(Z|\hat{Y},S)\leq H(Z)=1$, i.e., the maximum disparity is 1 bit.

**Example 1** (Pure Uniqueness). *Let $\hat{Y} = Z$ and $Z \perp\!\!\!\perp S$. The men ($Z = 1$) and women ($Z = 0$) are identically distributed across the clients. Suppose, the model only approves men but rejects women for a loan across both clients. This model is both locally and globally unfair, $\mathrm{I}(Z;\hat{Y}) = \mathrm{I}(Z;\hat{Y}|S) = 1$. This is a case of purely Unique Disparity since all the information about gender is derived exclusively from the model predictions; the client $S$ has no correlation with gender $Z$. Both Global and Local Disparities are in agreement. Here, $\mathrm{Uni}(Z{:}\hat{Y}|S) = 1$, $\mathrm{Red}(Z{:}\hat{Y},S) = 0$, and $\mathrm{Syn}(Z{:}\hat{Y},S) = 0$.*

**Example 2** (Pure Redundancy). *The client $S = 0$ has 90% women, while client $S = 1$ has 90% men. So, there is a correlation between the clients and gender. Suppose, the model approves everyone from client $S = 1$ while rejecting everyone in $S = 0$ (i.e., $\hat{Y} = S$). Such a model is locally fair because men and women are treated equally within a particular client, and $\mathrm{I}(Z;\hat{Y}|S) = 0$. However, the model is globally unfair since $\mathrm{I}(Z;\hat{Y}) = 0.53$. This is a case with pure Redundant Disparity since information about $Z$ is derived from both $\hat{Y}$ and $S$. Global and Local Disparities are in disagreement. Here, $\mathrm{Uni}(Z{:}\hat{Y}|S) = 0$, $\mathrm{Red}(Z{:}\hat{Y},S) = 0.53$, and $\mathrm{Syn}(Z{:}\hat{Y},S) = 0$.*

In general, pure Redundant Disparity is observed when $Z$ and $\hat{Y}$ are correlated and $Z - S - \hat{Y}$ form a *Markov chain*, i.e., $\hat{Y} = S$ and $S = g(Z)$ for some function $g$.

**Example 3** (Pure Synergy). *Let $\hat{Y} = Z \oplus S$ and $Z \perp\!\!\!\perp S$. The model approves men ($Z = 1$) from client $S = 0$ and women ($Z = 0$) from client $S = 1$, while others are rejected. Such a model is locally unfair, as it singularly prefers one gender within each client with $\mathrm{I}(Z;\hat{Y}|S) = 1$. However, it is globally fair since it maintains an equal approval rate for both men and women with $\mathrm{I}(Z;\hat{Y}) = 0$. This is a case with pure Masked Disparity as information about $Z$ that is not observable in either $\hat{Y}$ or $S$ individually is present jointly. Here, $\mathrm{Uni}(Z{:}\hat{Y}|S) = 0$, $\mathrm{Red}(Z{:}\hat{Y},S) = 0$, and $\mathrm{Syn}(Z{:}\hat{Y},S) = 1$.*

**Merits of PID.** These canonical examples demonstrate scenarios with pure uniqueness, redundancy, and synergy. In practice, there is usually a mixture of all of these disparities. i.e., non-zero Unique, Redundant, and Masked Disparities. In these scenarios, PID serves as a powerful tool that can disentangle the regions of agreement and disagreement between Local and Global Disparity, particularly when data is distributed non-identically across clients (also see experiments in Section 4). In contrast, traditional fairness metrics lack the granularity to capture these nuanced interactions, making PID an essential asset for a more comprehensive understanding and mitigation of disparities. Using PID, we can uncover the fundamental information-theoretic limits and trade-offs between Global and Local Disparities, which we will examine in greater depth next.

### 3.2 FUNDAMENTAL LIMITS ON TRADEOFFS BETWEEN LOCAL AND GLOBAL DISPARITY

We examine the use of local fairness to achieve global fairness, or scenarios where a model is trained to achieve local fairness and subsequently deployed at the global level. Since clients have direct

access only to their data, implementing local disparity mitigation techniques at the individual client level is both practical and convenient. Studies such as Cui et al. (2021) argue that local fairness is important as models are deployed at the local client level. However, this raises a critical question about the impact on global fairness. In Theorem 1, we formally demonstrate that even if local clients are able to use some optimal local mitigation methods and model aggregation techniques to achieve local fairness, the Global Disparity may still be greater than zero.

**Theorem 1** (Impossibility of Using Local Fairness to Attain Global Fairness). *As long as Redundant Disparity* $\text{Red}(Z{:}\hat{Y}, S) > 0$*, the Global Disparity* $\text{I}(Z; \hat{Y}) > 0$ *even if Local Disparity goes to* $0$*.*

In order to achieve local fairness, Unique and Masked Disparities must be reduced to zero. The proof leverages Proposition 1, particularly relying on non-negativity of Unique and Redundant Disparities (see Appendix D). Recall, Example 2 (Pure Redundancy), where the Local Disparity was zero, but the Global Disparity was $0.53$ as a result of the Redundant Disparity. This is not uncommon in real-world scenarios. Sensitive attributes like race or ethnicity may be correlated with location. For instance, one hospital may mainly cater to White patients, whereas another could predominantly serve Black patients. A model may be trained to achieve local fairness but would fail to be globally fair due to a non-zero Redundant Disparity, highlighting the region of disagreement (see Fig. 2).

We now consider the scenario where a model is able to achieve global fairness and is subsequently deployed at the local client level.

**Theorem 2** (Global Fairness Does Not Imply Local Fairness). *As long as Masked Disparity* $\text{Syn}(Z{:}\hat{Y}, S){>}0$*, local fairness will not be attained even if global fairness is attained.*

To achieve global fairness, the Unique and Redundant Disparities must reduce to zero. Recall Example 3 (Pure Synergy), where the model accepts men from client $S = 0$ and women from client $S = 1$, while rejecting all others. While this model is globally fair, it is not locally fair. This demonstrates that while it is possible to train a model to achieve global fairness, it may still exhibit disparity when deployed at the local level due to the canceling of disparities between clients. This effect is captured by the Masked Disparity.

We now discuss the necessary and sufficient conditions to achieve global fairness using local fairness.

**Theorem 3** (Necessary and Sufficient Condition to Achieve Global Fairness Using Local Fairness). *If Local Disparity* $\text{I}(Z; \hat{Y}|S)$ *goes to zero, then Global Disparity* $\text{I}(Z; \hat{Y})$ *also goes to zero, if and only if the Redundant Disparity* $\text{Red}(Z{:}\hat{Y}, S){=}0$*. A sufficient condition for* $\text{Red}(Z{:}\hat{Y}, S){=}0$ *is* $Z \perp\!\!\!\perp S$*.*

Theorem 3 suggests that if the sensitive attributes is uniformly distributed across clients the Redundant Disparity will reduce to zero (see proof in Appendix D). Hence, when the Local Disparity goes to zero, the Global Disparity will also decrease to zero. However, in practice, this proportion is fixed since the dataset at each client cannot be changed, i.e., $\text{I}(Z; S)$ is fixed. Therefore, we examine another more controllable condition to eliminate Redundant Disparity even when $\text{I}(Z; S) > 0$.

One might think that a potential solution to have $\text{Red}(Z{:}\hat{Y}, S) = 0$ is to enforce independence between $\hat{Y}$ and $S$, i.e., the model should make predictions at the same rate across all clients. However, interestingly, the PID literature demonstrates counterexamples (Kolchinsky, 2022) where this does not hold. We show that an additional condition of $\text{Syn}(Z{:}\hat{Y}, S) = 0$ is required.

**Lemma 3.** *A sufficient condition for* $\text{Red}(Z{:}\hat{Y}, S) = 0$ *is* $\text{Syn}(Z{:}\hat{Y}, S) = 0$ *and* $\hat{Y} \perp\!\!\!\perp S$*.*

**Remark 1.** *It is worth noting that the independence between* $\hat{Y}$ *and* $S$ *can be approximately achieved if the true* $Y$ *and* $S$ *are independent, as* $\hat{Y}$ *is an estimation of* $Y$*. The mutual information* $\text{I}(Y; S)$ *can provide insights into the anticipated value of* $\text{I}(\hat{Y}; S)$*, as FL typically aims to also achieve a reasonable level of accuracy. However, it is often the case that* $\text{I}(\hat{Y}; S)$ *is fixed due to the fixed nature of datasets at each client. It may even be possible to enforce* $\hat{Y} \perp\!\!\!\perp S$ *at the cost of accuracy.*

Lastly, we examine conditions to attain local fairness through global fairness.

**Theorem 4.** *Local disparity will always be less than Global Disparity if and only if Masked Disparity* $\text{Syn}(Z{:}\hat{Y}, S) = 0$*. A sufficient condition is when* $Z - \hat{Y} - S$ *form a Markov chain.*

**Remark 2** (Extension to Personalized Federated Learning Setting). *Interestingly, our results extend to the personalized FL setting, where client* $s$ *can tailor the final global model* $\hat{Y} = f(X)$ *into a*

*personalized version to improve local performance, $\hat{Y} = f_s(X)$. In this case, we can define the global model as a random variable $\hat{Y} = g(X, S)$ and all of the propositions would hold.*

### 3.3 AN OPTIMIZATION FRAMEWORK FOR EXPLORING THE ACCURACY FAIRNESS TRADE-OFF

We investigate the inherent trade-off between model accuracy and fairness in the FL context. We formulate the *Accuracy and Global-Local Fairness Optimality Problem (AGLFOP)*, an optimization to delineate the theoretical boundaries of accuracy and fairness trade-offs, capturing the optimal performance any model or FL technique can achieve for a specified dataset and client distribution.

Let $\Delta$ be the set of all joint distributions defined for $(Z, S, Y, \hat{Y})$. Let $\Delta_p$ be a set of all joint distributions $Q \in \Delta$ that maintain fixed marginals on $(Z, S, Y)$ as determined by a given dataset and client distribution, i.e., $\Delta_p = \{Q \in \Delta : \Pr_Q(Z{=}z, S{=}s, Y{=}y) = \Pr(Z{=}z, S{=}s, Y{=}y), \forall z, s, y\}$.

**Definition 4** (Accuracy and Global-Local Fairness Optimality Problem (AGLFOP)). *Let $\mathrm{I}_Q(Z; \hat{Y})$ and $\mathrm{I}_Q(Z; \hat{Y}|S)$ be Global and Local Disparity under distribution $Q$. Then, the AGLFOP for a specific dataset and client distribution is an optimization of the form:*

$$\underset{Q \in \Delta_p}{\arg\min} \quad err(Q) \quad \text{subject to} \quad \mathrm{I}_Q(Z; \hat{Y}) \le \epsilon_g, \quad \mathrm{I}_Q(Z; \hat{Y}|S) \le \epsilon_l, \tag{4}$$

*where $err(Q){=}\sum_{z,s,y,\hat{y}} \Pr_Q(Z{=}z, S{=}s, Y{=}y, \hat{Y}{=}\hat{y})\mathbb{I}(y \ne \hat{y})$, the classification error under distribution $Q$ ($\mathbb{I}(\cdot)$ denotes the indicator function). $err(Q) \in [0, 1]$ quantifies the proportion of incorrect predictions, calculated as the summation of the probabilities of misclassifying the true labels. The complement of the classification error, $1 - err(Q)$, quantifies the accuracy.*

**Theorem 5.** *The AGLFOP is a convex optimization problem.*

The AGLFOP is a convex optimization problem (see proof in Appendix E) that evaluates all potential joint distributions within the set $\Delta_p$ which includes the specific dataset and how they are distributed across clients. The true distribution of this given dataset across clients is $\Pr(Z = z, S = s, Y = y)$. This makes it an appropriate framework for investigating the accuracy-fairness trade-off. The Pareto front of this optimization problem facilitates a detailed study of the trade-offs, showcasing the maximum accuracy that can be attained for a given global and local fairness relaxation ($\epsilon_g, \epsilon_l$).

The set $\Delta_p$ can be further restricted for specialized applications, e.g., to constrain to all derived classifiers from an optimal classifier (Hardt et al., 2016). We can restrict our optimization space $\Delta_p$ to lie within the convex hull derived by the False Positive Rate (FPR) and True Positive Rate (TPR) of an initially trained classifier. This would characterize the accuracy-fairness tradeoff for all derived classifiers from the original trained classifier. The convex hall characterizes the distributions that can be achieved with any derived classifier. The constraints of AGLFOP can also be expressed using PID terms, offering intriguing insights that we will examine in the following section. The AGLFOP can be computed in any FL environment. Specifically, their computation necessitates the characterization of the joint distribution $\Pr(Z{=}z, S{=}s, Y{=}y) = \Pr(S{=}s)\Pr(Z{=}z|S{=}s)\Pr(Y{=}y|Z{=}z, S{=}s)$, which can be readily acquired by aggregating pertinent statistics across all participating clients.

**Remark 3** (Broader Potential). *The AGLFOP currently focuses on independence between $Z$ and $\hat{Y}$, but can be adapted to explore other fairness notions. It can also be used to study optimality in situations where different clients have varying fairness requirements, e.g., adhering to statistical parity globally while upholding equalized odds at the local level. Moreover, variants of this optimization problem can be developed to penalize only the worst-case client fairness scenarios.*

## 4 EXPERIMENTAL DEMONSTRATION

In this section, we provide experimental evaluations on synthetic and real-world datasets to validate our theoretical findings.[2] We investigate the PID of Global and Local Disparity under various conditions. We also examine the trade-offs between these fairness metrics and model accuracy.

**Data and Client Distribution.** We consider the following: *(1) Synthetic dataset:* A 2-D feature vector $X{=}(X_0, X_1)$ has a distribution, i.e., $X|_{Y=1} \sim \mathcal{N}((2, 2), \left[\begin{smallmatrix} 5 & 1 \\ 1 & 5 \end{smallmatrix}\right])$, $X|_{Y=0} \sim \mathcal{N}((-2, -2), \left[\begin{smallmatrix} 10 & 1 \\ 1 & 3 \end{smallmatrix}\right])$.

---

[2]Implementation is available at `https://github.com/FaisalHamman/FairFL-PID`

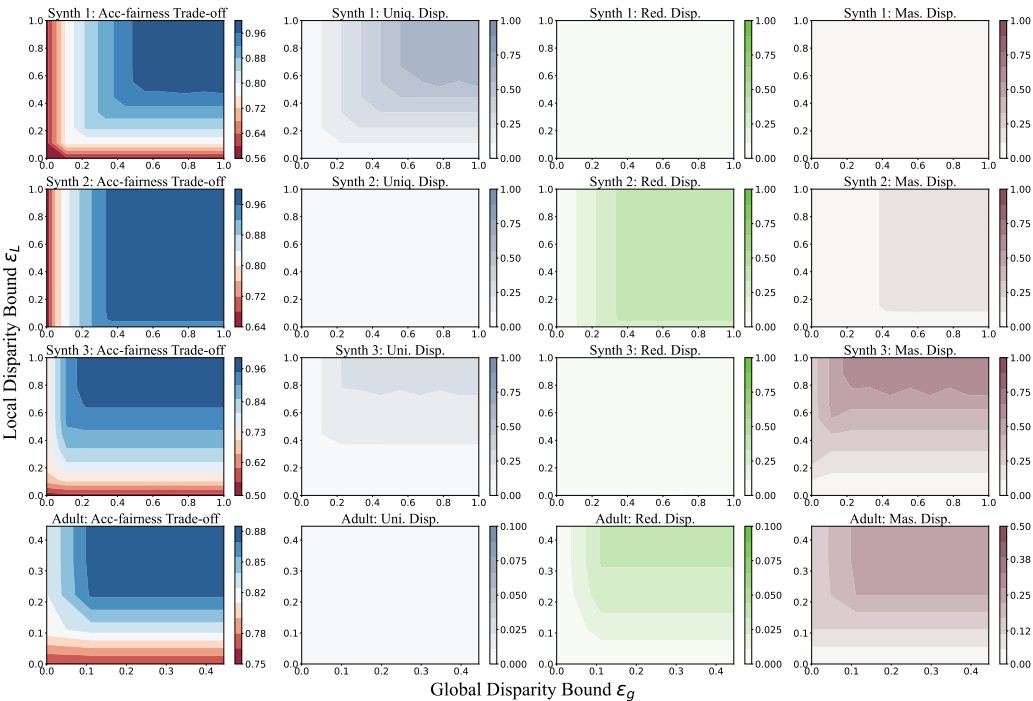

Figure 3: AGLFOP Pareto Frontiers for Synthetic and Adult Datasets with PID. (*first column*) shows maximum accuracy $(1 - err)$ that can be achieved on a dataset and client distribution for a given global and local fairness relaxation $(\epsilon_g, \epsilon_l)$. Synthetic data in scenario 1 (*first row*) is characterized by Unique Disparity. Local and global fairness agree, and accuracy trade-offs are balanced between them. Synthetic data in scenario 2 with $\alpha = 0.9$ (*second row*) is dominated by Redundant Disparity with trade-offs mainly between global fairness and accuracy (an accurate model could have zero Local Disparity but be globally unfair). Synthetic data in Scenario 3 (*third row*) is characterized by Masked Disparity with trade-offs mainly between local fairness and accuracy (an accurate model could have zero Global Disparity but be locally unfair). Adult data with heterogeneous split (*fourth row; details in Appendix F*), displaying predominantly Masked Disparity but notable presence of Redundant Disparity, capturing more complex relationships and trade-offs.

Assume binary sensitive attribute $Z=1$ if $X_0>0$ and $0$ otherwise to encode a dependence; and *(2) Adult dataset* (Dua & Graff, 2017) with gender as sensitive attribute. We consider three cases for partitioning the datasets across clients: (*Scenario* 1) sensitive-attribute independently distributed across clients, i.e., $Z \perp\!\!\!\perp S$, (*Scenario* 2) high sensitive-attribute heterogeneity across clients, i.e., $Z = S$ with probability $\alpha$, and (*Scenario* 3) high sensitive-attribute synergy level across clients, i.e., $Y = Z \oplus S$. Further details are described in Appendix F.

**Experiment A: Accuracy-Global-Local-Fairness Trade-off Pareto Front.** To study the trade-offs between model accuracy and different fairness constraints, we plot the Pareto frontiers for the AGLFOP. We solve for maximum accuracy $(1 - err)$ while varying global and local fairness relaxations $(\epsilon_g, \epsilon_l)$. We present results for synthetic and Adult datasets as well as PID terms for various data splitting scenarios across clients.[3] The three-way trade-off among accuracy, global, and local fairness can be visualized as a contour plot (see Fig. 3).

Interestingly, PID allows us to quantify the agreement and disagreement between local and global fairness. In scenarios characterized by Unique Disparity, local and global fairness agree, and accuracy trade-offs are balanced between them. In cases characterized by Redundant Disparity, the trade-off is primarily between accuracy and global fairness (the accuracy changes along the horizontal axis ($\epsilon_l$) seemingly nonexistent given ($\epsilon_g$)). In contrast, scenarios with Masked Disparity exhibit a trade-off that is primarily between accuracy and local fairness (the trade-off is across the vertical axis).

---

[3]We use Python *dit* package (James et al., 2018) for PID computation and *cvxpy* for convex solvers.

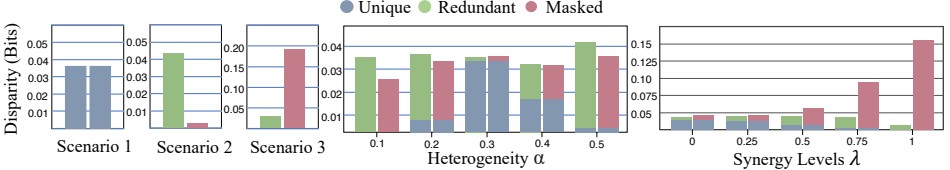

Figure 4: *(left)* Plot demonstrating scenarios with Unique, Redundant, and Masked Disparities for the Adult dataset (model trained using *FedAvg*). Unique Disparity when sensitive attributes are equally distributed across clients. Redundant Disparity when there is a dependency between clients and sensitive attributes (scenario 2; $\alpha = 0.9$). Masked Disparity is dominant with high sensitive attribute synergy level across clients. *(middle)* Illustrates PID for varying levels of sensitive attribute heterogeneity ($\alpha$; see details in Appendix F.2). When $\alpha$ is close to 0.3, the data is split evenly across clients (note $\Pr(Z{=}0){=}0.33$ for the Adult dataset), resulting in a higher level of Unique Disparity. As $\alpha$ deviates from 0.3, i.e., higher dependency between $Z$ and $S$, the Unique Disparity decreases while Redundant and Masked Disparity increases. *(right)* Illustrates relationship between the synergy level ($\lambda$; see details in Appendix F.2) and global and local fairness. As the synergy level increases, the Masked and Local Disparity increases as expected.

**Experiment B: Demonstrating Disparities in Federated Learning Settings.** In this experiment, we investigate the PID of disparities on the Adult dataset trained within a FL framework. We employ the *FedAvg* algorithm (McMahan et al., 2017) for training and analyze:

- *PID Across Various Splitting Scenarios.* We partition the dataset among clients based on Scenarios 1-3, utilize FedAvg for model training in each case, and examine the PID of both Local and Global Disparities (see Fig. 4). For each scenario, we also evaluate the effects of using a local disparity mitigation technique. This is achieved by incorporating a statistical parity regularizer at each client. The results and implementation details are presented in Table 1 in Appendix F.2.

- *PID Under Varying Sensitive Attribute Heterogeneity Level.* We partition the dataset across two clients with varying levels of sensitive attribute heterogeneity. We use $\alpha = \Pr(Z = 0 | S = 0)$ to control the sensitive attribute heterogeneity level across clients. Our results are summarized in Fig. 4 and Table 2 in Appendix F.2.

- *Observing Levels of Masked Disparity.* We partition the dataset with varying sensitive attribute synergy levels across clients to study the impact on the Masked Disparity. The *synergy level* $\lambda \in [0, 1]$ measures how closely the true label $Y$ aligns with $Z \oplus S$ (see Definition 10 in Appendix F.2). Results are summarized in Fig. 4 and Table 3 in Appendix F.2.

- *Experiments Involving Multiple Clients.* We experiment with multiple clients $K = 5$ and $K = 10$. Our findings are presented in Fig. 6, Fig. 5 and Table 4 in Appendix F.2.

**Discussion.** Our information-theoretic framework provides a nuanced understanding of the sources of disparity in FL, namely, Unique, Redundant, and Masked disparity. Our experiments offer insights into the agreement and disagreement between local and global fairness under various data distributions. Our experiments and theoretical results show that depending on the data distribution achieving one can often come at the cost of the other (disagreement). The nature of the data distribution across clients significantly impacts the disparity that dominates. Our optimization framework establishes the accuracy-fairness tradeoffs for a dataset and client distribution.

Importantly, our research can: *(i)* inform the use of local disparity mitigation techniques and their convergence and effectiveness when deployed in practice; and *(ii)* also serve as a valuable tool for policy decision-making, shedding light on the effects of model bias at both the global and local levels. This is particularly relevant in the expanding field of algorithmic fairness auditing (Hamman et al., 2023; Yan & Zhang, 2022). Future studies could also investigate how this approach could be extended to more sophisticated fairness measures. The estimation of PID terms largely depends on *(i)* the empirical estimators of the probability distributions; and *(ii)* the efficiency of the convex optimization algorithm used for calculating the unique information. As the number of clients or sensitive attributes increases, the computational cost may rise accordingly. Future work could explore alternative efficient PID computation techniques (Belghazi et al., 2018; Venkatesh & Schamberg, 2022; Pakman et al., 2021; Kleinman et al., 2021).

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

## A    RELEVANT BACKGROUND ON INFORMATION THEORETIC MEASURES

We outline key information-theoretic measures pertinent to this paper's discussions.

**Definition 5** (Entropy). *Entropy quantifies the uncertainty or unpredictability of a random variable $Z$. It is mathematically defined by the equation:*

$$H(Z) = -\sum_z \Pr(Z = z) \log \Pr(Z = z). \tag{5}$$

**Definition 6** (Mutual Information). *Mutual Information, $\mathrm{I}(Z; \hat{Y})$, quantifies the amount of information obtained about random variable $Z$ through $\hat{Y}$. Specifically, it measures the degree of dependence between two variables, $Z$ and $\hat{Y}$, capturing both linear and non-linear dependencies:*

$$\mathrm{I}(Z; \hat{Y}) = \sum_{z,\hat{y}} \Pr(Z = z, \hat{Y} = \hat{y}) \log \frac{\Pr(Z = z, \hat{Y} = \hat{y})}{\Pr(Z = z) \Pr(\hat{Y} = \hat{y})}. \tag{6}$$

**Definition 7** (Conditional Mutual Information). *The conditional mutual information, $I(Z;\hat{Y}|S)$, measures the dependency between $Z$ and $\hat{Y}$, conditioned on $S$:*

$$\mathrm{I}(Z;\hat{Y}|S) = \sum_{s,z,\hat{y}} \Pr(S=s, Z=z, \hat{Y}=\hat{y}) \log \frac{\Pr(Z=z, \hat{Y}=\hat{y}|S=s)}{\Pr(Z=z|S=s)\Pr(\hat{Y}=\hat{y}|S=s)}, \quad (7)$$

*or alternatively,*

$$\mathrm{I}(Z;\hat{Y}|S) = \sum_s \Pr(S=s)\mathrm{I}(Z;\hat{Y}|S=s). \quad (8)$$

**Mutual Information as a Measure of Fairness.** Mutual information can be used as a measure of the unfairness or disparity of a model. Mutual Information has been interpreted as the dependence between sensitive attribute $Z$ and model prediction $\hat{Y}$ (captures correlation as well as all non-linear dependencies). Mutual information is zero if and only if $Z$ and $\hat{Y}$ are independent. This means that if the model's predictions are highly correlated with sensitive attributes (like gender or race), that's a sign of unfairness. Mutual information has been explored in fairness in the context of centralized machine learning in Kamishima et al. (2011); Cho et al. (2020); Kang et al. (2021).

In recent work, Venkatesh et al. (2021) provides another interpretation of mutual information $\mathrm{I}(Z;\hat{Y})$ in fairness as the accuracy of predicting $Z$ from $\hat{Y}$ (or the expected probability of error in correctly guessing $Z$ from $\hat{Y}$) from Fano's inequality. Even in information bottleneck literature, mutual information has been interpreted as a measure of how well one random variable predicts (or, aligns with) the other (Goldfeld & Polyanskiy, 2020).

For local fairness, we are interested in the dependence between model prediction $\hat{Y}$ and sensitive attributes $Z$ at each and every client, i.e., the dependence between $\hat{Y}$ and $Z$ conditioned on the client S. For example, the disparity at client $S=1$ is $\mathrm{I}(Z;\hat{Y}|S=1)$ (the mutual information (dependence) between model prediction and sensitive attribute conditioned on client $S=1$ (considering data at client $S=1$). Our measure for Local Disparity is the conditional mutual information (dependence) between $Z$ and $\hat{Y}$ conditioned on $S$, denoted as $\mathrm{I}(Z;\hat{Y}|S)$. Local disparity $\mathrm{I}(Z;\hat{Y}|S) = \sum_s p(s)I(Z;\hat{Y}|S=s)$, is an average of the disparity at each client weighted by the $p(s)$, the proportion of data at client $S=s$. The Local Disparity is zero if and only if all client has zero disparity in their local dataset.

## B  ADDITIONAL RESULTS AND PROOFS FOR SECTION 3

**Lemma 1** (Relationship between Global Statistical Parity Gap and $\mathrm{I}(Z;\hat{Y})$). *Let $\Pr(Z=0) = \alpha$. The gap $SP_{global} = |\Pr(\hat{Y}=1|Z=1) - \Pr(\hat{Y}=1|Z=0)|$ is bounded by $\frac{\sqrt{0.5\,\mathrm{I}(Z;\hat{Y})}}{2\alpha(1-\alpha)}$.*

*Proof.* Mutual information can be expressed as KL divergence:

$$\mathrm{I}(Z;\hat{Y}) = D_{KL}\left(\Pr(\hat{Y}, Z)||\Pr(\hat{Y})\Pr(Z)\right). \quad (9)$$

Using Pinsker's Inequality (Canonne, 2022),

$$d_{TV}(Q_1, Q_2) \le \sqrt{0.5 D_{KL}(Q_1||Q_2)} \quad (10)$$

where, $d_{TV}(Q_1, Q_2)$ is the total variation between two probability distributions $Q_1, Q_2$.

$$d_{TV}\left(\Pr(\hat{Y}, Z), \Pr(\hat{Y})\Pr(Z)\right) = \frac{1}{2}\sum_{\hat{y},z}\left|\Pr(\hat{Y} = \hat{y}, Z = z) - \Pr(\hat{Y} = \hat{y})\Pr(Z = z)\right|$$

$$= \sum_z \Pr(Z = z)\sum_{\hat{y},z}\frac{1}{2}\left|\Pr(\hat{Y} = \hat{y}|Z = z) - \Pr(\hat{Y} = \hat{y})\right|$$

$$= \frac{1}{2}\Pr(Z = 1)\left[|\Pr(\hat{Y} = 1|Z = 1) - \Pr(\hat{Y} = 1)| + |\Pr(\hat{Y} = 0|Z = 1) - \Pr(\hat{Y} = 0)|\right]$$

$$+ \frac{1}{2}\Pr(Z = 0)\left[|\Pr(\hat{Y} = 1|Z = 0) - \Pr(\hat{Y} = 1)| + |\Pr(\hat{Y} = 0|Z = 0) - \Pr(\hat{Y} = 0)|\right]$$

$$= \frac{1}{2}\alpha(1 - \alpha)|SP1| + \frac{1}{2}\alpha(1 - \alpha)|SP0| + \frac{1}{2}\alpha(1 - \alpha)|SP1| + \frac{1}{2}\alpha(1 - \alpha)|SP0|$$

$$= \alpha(1 - \alpha)|SP1| + \alpha(1 - \alpha)|SP0| \tag{11}$$

where $\Pr(Z = 0) = 1 - \Pr(Z = 1) = \alpha$, and

$$SPi = \Pr(\hat{Y} = i|Z = 1) - \Pr(\hat{Y} = i|Z = 0) = \Pr(\hat{Y} = i|Z = 1) - \Pr(\hat{Y} = i).$$

To complete the proof, we show:

$$SP1 = \Pr(\hat{Y} = 1|Z = 1) - \Pr(\hat{Y} = 1)$$

$$= \Pr(\hat{Y} = 1|Z = 1) - \left(1 - \Pr(\hat{Y} = 0)\right)$$

$$= -1 + \Pr(\hat{Y} = 1|Z = 1) + \Pr(\hat{Y} = 0)$$

$$= -\Pr(\hat{Y} = 0|Z = 1) + \Pr(\hat{Y} = 0) = -SP0.$$

Hence, $|SP1| = |SP0|$. From Equation 11 we have: $2\alpha(1 - \alpha)|SP_1| \leq \sqrt{0.5\mathrm{I}(Z; \hat{Y})}$.

**Remark 4** (Tightness of Lemma 1). *Since our proof exclusively utilizes Pinsker's inequality, their tightness is equivalent. Given $\mathrm{I}(Z; \hat{Y}) \leq \min\{H(Z), H(\hat{Y})\} \leq H(\hat{Y})$ and $H(Y) \leq 1$ in binary classification. Hence, $\mathrm{I}(Z; \hat{Y}) \leq 1$ which is aligned with the known tight regime of Pinsker's inequality (i.e., $D_{KL}(P||Q) \leq 1$) (Canonne, 2022). The inequality is tighter with smaller mutual information $\mathrm{I}(Z; \hat{Y})$ values.*

$\square$

**Lemma 2.** $\mathrm{I}(Z; \hat{Y}|S) = 0$ *if and only if* $\Pr(\hat{Y}{=}1|Z{=}1, S{=}s) = \Pr(\hat{Y}{=}1|Z{=}0, S{=}s)$ *at all clients $s$.*

*Proof.* We aim to establish that $\mathrm{I}(Z; \hat{Y}|S) = 0$ if and only if $\Pr(\hat{Y} = 1|Z = 1, S = s) = \Pr(\hat{Y} = 1|Z = 0, S = s)$ for all clients $s$. For brevity, we denote $\Pr(Z = z, S = s, Y = y) = p(z, s, y)$.

*Forward Direction*: Assume $\mathrm{I}(Z; \hat{Y}|S) = 0$.

From Definition 7, we have:

$$\mathrm{I}(Z; \hat{Y}|S) = \sum_{s,z,\hat{y}} p(s, z, \hat{y}) \log\left(\frac{p(z, \hat{y}|s)}{p(z|s)p(\hat{y}|s)}\right) = 0.$$

This implies that $\log\left(\frac{p(z,\hat{y}|s)}{p(z|s)p(\hat{y}|s)}\right) = 0$ for all $s, z, \hat{y}$, and consequently $\frac{p(z,\hat{y}|s)}{p(z|s)p(\hat{y}|s)} = 1 \ \forall s$.

Observing that $p(z, \hat{y}|s) = p(z|s)p(\hat{y}|z, s)$, we deduce that $\frac{p(z|s)p(\hat{y}|z,s)}{p(z|s)p(\hat{y}|s)} = 1$.

From this, it directly follows that $p(\hat{y}|z, s) = p(\hat{y}|s)$, and thus $\Pr(\hat{Y} = 1|Z = 1, S = s) = \Pr(\hat{Y} = 1|Z = 0, S = s)$.

*Reverse Direction*: Assume $\Pr(\hat{Y} = 1|Z = 1, S = s) = \Pr(\hat{Y} = 1|Z = 0, S = s)$ for all $s$.

This implies $p(\hat{y}|s, z) = p(\hat{y}|s)$ for all $s, z, \hat{y}$. Plugging this into the definition of conditional mutual information, we find $\mathrm{I}(Z; \hat{Y}|S) = 0$.

Thus, both directions of the equivalence are proven, concluding the proof. □

**Corollary 1.** *The statistical parity at each client $s$ can be expressed as*

$$|SP_s| \leq \frac{\sqrt{0.5 \, \mathrm{I}(Z; \hat{Y}|S = s)}}{2\alpha_s(1 - \alpha_s)}$$

*where, $\alpha_s = \Pr(Z = 0|S = s) = 1 - \Pr(Z = 1|S = s)$.*

**Definition 8** (Difference Between Local and Global Disparity). *The difference between Global and Local Disparity is: $\mathrm{I}(Z; \hat{Y}) - \mathrm{I}(Z; \hat{Y}|S) = \mathrm{I}(Z; \hat{Y}; S)$. This term is the "interaction information," which, unlike other mutual-information-based measures, can be positive or negative.*

Interaction information quantifies the redundancy and synergy present in a system. In FL, positive interaction information indicates a system with high levels of redundancy and Global Disparity, while negative interaction information indicates a system with high levels of synergy and Local Disparity. Interaction information can inform the trade-off between Local and Global Disparity.

## C   PROOF FOR SECTION 3.1

**Proposition 1.** *The Global and Local Disparity in FL can be decomposed into non-negative terms:*

$$\mathrm{I}(Z; \hat{Y}) = \mathrm{Uni}(Z{:}\hat{Y}|S) + \mathrm{Red}(Z{:}\hat{Y}, S). \tag{2}$$

$$\mathrm{I}(Z; \hat{Y}|S) = \mathrm{Uni}(Z{:}\hat{Y}|S) + \mathrm{Syn}(Z{:}\hat{Y}, S). \tag{3}$$

*Proof.* Equation 2 follows directly from the PID terms definition.

$$\mathrm{Uni}(Z{:}\hat{Y}|S) + \mathrm{Red}(Z{:}\hat{Y}, S) = \min_{Q \in \Delta_p} \mathrm{I}_Q(Z; \hat{Y}|S) + \mathrm{I}(Z; \hat{Y}) - \min_{Q \in \Delta_p} \mathrm{I}_Q(Z; \hat{Y}|S) = \mathrm{I}(Z; \hat{Y}).$$

Equation 3 follows from PID terms definition and the chain rule of mutual information.

$$\begin{aligned} \mathrm{Uni}(Z{:}\hat{Y}|S) + \mathrm{Syn}(Z{:}\hat{Y}, S) &= \min_{Q \in \Delta_p} \mathrm{I}_Q(Z; \hat{Y}|S) + \mathrm{I}(Z; \hat{Y}, S) - \mathrm{I}(Z; S) - \min_{Q \in \Delta_p} \mathrm{I}_Q(Z; \hat{Y}|S) \\ &= \mathrm{I}(Z; S) + \mathrm{I}(Z; \hat{Y}|S) - \mathrm{I}(Z; S) \\ &= \mathrm{I}(Z; \hat{Y}|S). \end{aligned}$$

Now, we prove the non-negativity property of PID decomposition.

$\mathrm{Uni}(Z{:}\hat{Y}|S) = \min_{Q \in \Delta_p} \mathrm{I}_Q(Z; \hat{Y}|S)$ is non-negative since the conditional mutual information is non-negative by definition.

$\mathrm{Syn}(Z{:}\hat{Y}, S) = \mathrm{I}(Z; \hat{Y}|S) - \min_{Q \in \Delta_p} \mathrm{I}_Q(Z; \hat{Y}|S) \geq \mathrm{I}(Z; \hat{Y}|S) - \mathrm{I}(Z; \hat{Y}|S) = 0$

The Redundant Disparity:

$$\mathrm{Red}(Z{:}\hat{Y}, S) = \mathrm{I}(Z; \hat{Y}) - \min_{Q \in \Delta_p} \mathrm{I}_Q(Z; \hat{Y}|S) = \max_{Q \in \Delta_p} \mathrm{I}_Q(\hat{Y}; Z) - \mathrm{I}_Q(Z; \hat{Y}|S)$$

First equality holds by definition. Second equality holds since marginals on $(\hat{Y}, Z)$ is fixed in $\Delta_p$, hence, $\max_{Q \in \Delta_p} \mathrm{I}_Q(\hat{Y}; Z) = \mathrm{I}(\hat{Y}; Z)$.

To prove non-negativity of redundant disparity, we construct a distribution $Q_0$ such that:

$$\Pr_{Q_0}(Z = z, \hat{Y} = y, S = s) = \frac{\Pr(Z = z, \hat{Y} = y) \Pr(Z = z, S = s)}{\Pr(Z = z)}$$

Next, we show $Q_0 \in \Delta_p$. Recall the set $\Delta_p$ in Definition 1:

$$\Delta_p = \{Q \in \Delta : \Pr_Q(Z = z, \hat{Y} = y) = \Pr(Z = z, \hat{Y} = y), \Pr_Q(Z = z, S = s) = \Pr(Z = z, S = s)\}.$$

$$\Pr_{Q_0}(Z = z, \hat{Y} = y) = \sum_s \Pr_{Q_0}(Z = z, \hat{Y} = y, S = s) = \sum_s \frac{\Pr(Z = z, \hat{Y} = y)}{\Pr(Z = z)} \Pr(Z = z, S = s)$$

$$= \frac{\Pr(Z = z, \hat{Y} = y)}{\Pr(Z = z)} \sum_s \Pr(Z = z, S = s) = \Pr(Z = z, \hat{Y} = y).$$

$$\Pr_{Q_0}(Z = z, S = s) = \sum_{\hat{y}} \Pr_{Q_0}(Z = z, \hat{Y} = y, S = s) = \sum_{\hat{y}} \frac{\Pr(Z = z, \hat{Y} = y)\Pr(Z = z, S = s)}{\Pr(Z = z)}$$

$$= \frac{\Pr(Z = z, S = s)}{\Pr(Z = z)} \sum_{\hat{y}} \Pr(Z = z, \hat{Y} = y) = \Pr(Z = z, S = s).$$

Marginals of $Q_0$ satisfy conditions on set $\Delta_p$, hence $Q_0 \in \Delta_p$. Also, note that by construction of $Q_0$, $\hat{Y}$ and $S$ are independent conditioned on $Z$, i.e., $I_{Q_0}(\hat{Y}; S|Z) = 0$. Hence, we have:

$$\begin{aligned}
\mathrm{Red}(Z{:}\hat{Y}, S) &\overset{(a)}{=} \max_{Q \in \Delta_p} I_Q(Z; \hat{Y}) - I_Q(Z; \hat{Y}|S) \\
&\overset{(b)}{\geq} I_{Q_0}(Z; \hat{Y}) - I_{Q_0}(Z; \hat{Y}|S) \\
&\overset{(c)}{=} H_{Q_0}(Z) + H_{Q_0}(\hat{Y}) - H_{Q_0}(Z, \hat{Y}) - H_{Q_0}(Z|S) - H_{Q_0}(\hat{Y}|S) + H_{Q_0}(Z, \hat{Y}|S) \\
&\overset{(d)}{=} I_{Q_0}(\hat{Y}; S) - I_{Q_0}(\hat{Y}; S|Z) \\
&\overset{(e)}{=} I_{Q_0}(\hat{Y}; S) \overset{(f)}{\geq} 0.
\end{aligned}$$

Here, $(a)$ hold from definition of $\mathrm{Red}(Z{:}\hat{Y}, S)$, $(b)$ hold since $Q_0 \in \Delta_p$, $(c)$-$(d)$ holds from expressing mutual information in terms of entropy, $(e)$ hold since $I_{Q_0}(\hat{Y}; S|Z) = 0$, $(f)$ holds from non-negativity property of mutual information.

$\square$

## D  ADDITIONAL RESULTS AND PROOFS FOR SECTION 3.2

**Theorem 1** (Impossibility of Using Local Fairness to Attain Global Fairness). *As long as Redundant Disparity* $\mathrm{Red}(Z{:}\hat{Y}, S) > 0$*, the Global Disparity* $I(Z; \hat{Y}) > 0$ *even if Local Disparity goes to* $0$.

*Proof.* For completeness, we have provided a detailed proof that demonstrates the non-negativity property of the terms involved.

$\mathrm{Uni}(Z{:}\hat{Y}|S) = \min_{Q \in \Delta_p} I_Q(Z; \hat{Y}|S)$ is non-negative since the conditional mutual information is non-negative by definition.

$\mathrm{Syn}(Z{:}\hat{Y}, S) = I(Z; \hat{Y}|S) - \min_{Q \in \Delta_p} I_Q(Z; \hat{Y}|S) \geq I(Z; \hat{Y}|S) - I(Z; \hat{Y}|S) = 0.$

The Redundant Disparity:

$$\mathrm{Red}(Z{:}\hat{Y}, S) = I(Z; \hat{Y}) - \min_{Q \in \Delta_p} I_Q(Z; \hat{Y}|S) = \max_{Q \in \Delta_p} I_Q(\hat{Y}; Z) - I_Q(Z; \hat{Y}|S)$$

First equality holds by definition. Second equality holds since marginals on $(\hat{Y}, Z)$ is fixed in $\Delta_p$, hence, $\max_{Q \in \Delta_p} I_Q(\hat{Y}; Z) = I(\hat{Y}; Z)$.

To prove non-negativity of redundant disparity, we construct a distribution $Q_0$ such that:

$$\Pr_{Q_0}(Z = z, \hat{Y} = y, S = s) = \frac{\Pr(Z = z, \hat{Y} = y)\Pr(Z = z, S = s)}{\Pr(Z = z)}$$

Next, we show $Q_0 \in \Delta_p$. Recall the set $\Delta_p$ in Definition 1:

$$\Delta_p = \{Q \in \Delta : \Pr_Q(Z = z, \hat{Y} = y) = \Pr(Z = z, \hat{Y} = y), \Pr_Q(Z = z, S = s) = \Pr(Z = z, S = s)\}.$$

$$\Pr_{Q_0}(Z = z, \hat{Y} = y) = \sum_s \Pr_{Q_0}(Z = z, \hat{Y} = y, S = s) = \sum_s \frac{\Pr(Z = z, \hat{Y} = y)}{\Pr(Z = z)} \Pr(Z = z, S = s)$$

$$= \frac{\Pr(Z = z, \hat{Y} = y)}{\Pr(Z = z)} \sum_s \Pr(Z = z, S = s) = \Pr(Z = z, \hat{Y} = y).$$

$$\Pr_{Q_0}(Z = z, S = s) = \sum_{\hat{y}} \Pr_{Q_0}(Z = z, \hat{Y} = y, S = s) = \sum_{\hat{y}} \frac{\Pr(Z = z, \hat{Y} = y) \Pr(Z = z, S = s)}{\Pr(Z = z)}$$

$$= \frac{\Pr(Z = z, S = s)}{\Pr(Z = z)} \sum_{\hat{y}} \Pr(Z = z, \hat{Y} = y) = \Pr(Z = z, S = s).$$

Marginals of $Q_0$ satisfy conditions on set $\Delta_p$, hence $Q_0 \in \Delta_p$. Also, note that by construction of $Q_0$, $\hat{Y}$ and $S$ are independent conditioned on $Z$, i.e., $\mathrm{I}_{Q_0}(\hat{Y}; S|Z) = 0$. Hence, we have:

$$\mathrm{Red}(Z{:}\hat{Y}, S) \stackrel{(a)}{=} \max_{Q \in \Delta_p} \mathrm{I}_Q(Z; \hat{Y}) - \mathrm{I}_Q(Z; \hat{Y}|S)$$

$$\stackrel{(b)}{\geq} \mathrm{I}_{Q_0}(Z; \hat{Y}) - \mathrm{I}_{Q_0}(Z; \hat{Y}|S)$$

$$\stackrel{(c)}{=} H_{Q_0}(Z) + H_{Q_0}(\hat{Y}) - H_{Q_0}(Z, \hat{Y}) - H_{Q_0}(Z|S) - H_{Q_0}(\hat{Y}|S) + H_{Q_0}(Z, \hat{Y}|S)$$

$$\stackrel{(d)}{=} \mathrm{I}_{Q_0}(\hat{Y}; S) - \mathrm{I}_{Q_0}(\hat{Y}; S|Z)$$

$$\stackrel{(e)}{=} \mathrm{I}_{Q_0}(\hat{Y}; S) \stackrel{(f)}{\geq} 0.$$

Here, $(a)$ hold from definition of $\mathrm{Red}(Z{:}\hat{Y}, S)$, $(b)$ hold since $Q_0 \in \Delta_p$, $(c)$-$(d)$ holds from expressing mutual information in terms of entropy, $(e)$ hold since $\mathrm{I}_{Q_0}(\hat{Y}; S|Z) = 0$, $(f)$ holds from non-negativity property of mutual information.

Hence, from proposition 1, we prove Theorem 1.

As Local Disparity $\mathrm{I}(Z; \hat{Y}|S) \to 0$, then $\mathrm{Uni}(Z{:}\hat{Y}|S) \to 0$ and $\mathrm{Syn}(Z{:}\hat{Y}, S) \to 0$, therefore the Global Disparity $\mathrm{I}(Z; \hat{Y}) \to \mathrm{Red}(Z{:}\hat{Y}, S) \geq 0$. $\square$

**Theorem 2** (Global Fairness Does Not Imply Local Fairness). *As long as Masked Disparity* $\mathrm{Syn}(Z{:}\hat{Y}, S) > 0$, *local fairness will not be attained even if global fairness is attained.*

*Proof.* Proof requires the non-negativity property of PID terms (follows similarly from proof of Theorem 1). The argument then goes as follows:

As Global Disparity $\mathrm{I}(Z; \hat{Y}) \to 0$, then $\mathrm{Uni}(Z{:}\hat{Y}|S) \to 0$ and $\mathrm{Red}(Z{:}\hat{Y}, S) \to 0$, therefore the Local Disparity $\mathrm{I}(Z; \hat{Y}|S) \to \mathrm{Syn}(Z{:}\hat{Y}, S) \geq 0$. $\square$

**Theorem 3** (Necessary and Sufficient Condition to Achieve Global Fairness Using Local Fairness). *If Local Disparity* $\mathrm{I}(Z; \hat{Y}|S)$ *goes to zero, then Global Disparity* $\mathrm{I}(Z; \hat{Y})$ *also goes to zero, if and only if the Redundant Disparity* $\mathrm{Red}(Z{:}\hat{Y}, S) = 0$. *A sufficient condition for* $\mathrm{Red}(Z{:}\hat{Y}, S) = 0$ *is* $Z \perp\!\!\!\perp S$.

*Proof.* From the PID of Local and Global Disparity,

$$\mathrm{I}(Z; \hat{Y}) = \mathrm{Uni}(Z{:}\hat{Y}|S) + \mathrm{Red}(Z{:}\hat{Y}, S),$$

$$\mathrm{I}(Z; \hat{Y}|S) = \mathrm{Uni}(Z{:}\hat{Y}|S) + \mathrm{Syn}(Z{:}\hat{Y}, S).$$

Therefore if, $\mathrm{I}(Z; \hat{Y}|S) = 0$, then $\mathrm{Uni}(Z{:}\hat{Y}|S) = 0$. Hence,

$$\mathrm{I}(Z; \hat{Y}) = \mathrm{Red}(Z{:}\hat{Y}, S)$$

$$\mathrm{I}(Z; \hat{Y}) = 0 \iff \mathrm{Red}(Z{:}\hat{Y}, S) = 0.$$

To prove the sufficient condition, we leverage the PID of $\mathrm{I}(Z; S)$ and the non-negative property of the PID terms:

$$\mathrm{I}(Z; S) = \mathrm{Uni}(Z{:}S|\hat{Y}) + \mathrm{Red}(Z{:}\hat{Y}, S)$$

$$\mathrm{I}(Z; S) \geq \mathrm{Red}(Z{:}\hat{Y}, S).$$

Hence, $Z \perp\!\!\!\perp S \implies \mathrm{Red}(Z{:}\hat{Y}, S) = 0.$ $\qquad\square$

**Lemma 3.** *A sufficient condition for* $\mathrm{Red}(Z{:}\hat{Y}, S) = 0$ *is* $\mathrm{Syn}(Z{:}\hat{Y}, S) = 0$ *and* $\hat{Y} \perp\!\!\!\perp S$.

*Proof.* Interaction information expressed in PID terms (see Definition 8):

$$\mathrm{I}(Z; \hat{Y}; S) = \mathrm{I}(Z; \hat{Y}) - \mathrm{I}(Z; \hat{Y}|S) = \mathrm{Red}(Z{:}\hat{Y}, S) - \mathrm{Syn}(Z; \hat{Y}, S).$$

If Masked Disparity $\mathrm{Syn}(Z; \hat{Y}, S) = 0$, we have:

$$\mathrm{I}(Z; \hat{Y}; S) = \mathrm{I}(Z; \hat{Y}) - \mathrm{I}(Z; \hat{Y}|S) = \mathrm{Red}(Z{:}\hat{Y}, S) \geq 0$$

Since the interaction information is positive and symmetric,

$$\mathrm{I}(\hat{Y}; S) \geq \mathrm{I}(\hat{Y}; S) - \mathrm{I}(\hat{Y}; S|Z) = \mathrm{Red}(Z{:}\hat{Y}, S).$$

Therefore, $\hat{Y} \perp\!\!\!\perp S \implies \mathrm{Red}(Z{:}\hat{Y}, S) = 0.$ $\qquad\square$

**Theorem 4.** *Local disparity will always be less than Global Disparity if and only if Masked Disparity* $\mathrm{Syn}(Z{:}\hat{Y}, S) = 0$. *A sufficient condition is when* $Z - \hat{Y} - S$ *form a Markov chain.*

*Proof.* By leveraging the PID of $\mathrm{I}(Z; S|\hat{Y})$,

$$\mathrm{I}(Z; S|\hat{Y}) = \mathrm{Uni}(Z{:}S|\hat{Y}) + \mathrm{Syn}(Z{:}\hat{Y}, S).$$

Markov chain $Z - \hat{Y} - S$ implies, $\mathrm{I}(Z; S|\hat{Y}) = 0$. Hence, $\mathrm{Syn}(Z{:}\hat{Y}, S) = 0$.

Rest of proof follows from nonnegative property of PID terms:

$$\mathrm{I}(Z; \hat{Y}|S) = \mathrm{Uni}(Z{:}\hat{Y}|S) \leq \mathrm{Uni}(Z{:}\hat{Y}|S) + \mathrm{Red}(Z{:}\hat{Y}, S) = \mathrm{I}(Z; \hat{Y}).$$

$\qquad\square$

# E    PROOFS FOR SECTION 3.3

**Definition 9** (Convex Function). *A function* $f : \mathbb{R}^n \to \mathbb{R}$ *is said to be convex if, for all* $x_1, x_2 \in \mathbb{R}^n$ *and for all* $\lambda \in [0, 1]$, *the following inequality holds:*

$$f(\lambda x_1 + (1 - \lambda)x_2) \leq \lambda f(x_1) + (1 - \lambda)f(x_2). \qquad (12)$$

**Lemma 4** (Log Sum Inequality). *The log-sum inequality states that for any two sequences of non-negative numbers* $a_1, a_2, \ldots, a_n$ *and* $b_1, b_2, \ldots, b_n$, *the following inequality holds:*

$$\sum_{i=1}^{n} a_i \log\left(\frac{a_i}{b_i}\right) \geq \left(\sum_{i=1}^{n} a_i\right) \log\left(\frac{\sum_{i=1}^{n} a_i}{\sum_{i=1}^{n} b_i}\right). \qquad (13)$$

**Theorem 5.** *The AGLFOP is a convex optimization problem.*

*Proof.* The set $\Delta_p$ is a convex set, since for any two points $Q_1, Q_2 \in \Delta_p$, their convex combination also lies in $\Delta_p$ (probability simplex). To prove AGFOP is a convex optimization problem, we show each term is convex in $Q$ using the definition of a convex function (see Definition 9).

Let $Q \in \Delta_p$ denote the joint distribution for $(Z, S, Y, \hat{Y})$. For brevity, we denote $\Pr(Z = z, S = s, Y = y) = p(z, s, y)$, the fixed marginals on $(Z, S, Y)$. Additionally, we denote $\Pr_Q(Z = z, S = s, Y = y, \hat{Y} = \hat{y}) = Q(z, s, y, \hat{y})$ as the probability under distribution Q.

To prove that $err(Q)$ is convex, we need to show: $err(Q_\lambda) \leq \lambda err(Q_1) + (1-\lambda)err(Q_2)$, where $Q_\lambda = \lambda Q_1 + (1-\lambda)Q_2$, $\forall Q_1, Q_2 \in \Delta_p$, and $\lambda \in [0,1]$.

We first express $err(Q_\lambda)$ as:

$$
\begin{aligned}
err(Q_\lambda) &= \sum_{z,s,y,\hat{y}} Q_\lambda(z,s,y,\hat{y}) \cdot \mathbb{I}(y \neq \hat{y}) \\
&= \lambda \sum_{z,s,y,\hat{y}} Q_1(z,s,y,\hat{y}) \cdot \mathbb{I}(y \neq \hat{y}) + (1-\lambda) \sum_{z,s,y,\hat{y}} Q_2(z,s,y,\hat{y}) \cdot \mathbb{I}(y \neq \hat{y}) \\
&= \lambda err(Q_1) + (1-\lambda)err(Q_2).
\end{aligned}
$$

Thus, $err(Q)$ is convex as it satisfies the convexity condition.

To prove convexity of $\mathrm{I}_Q(Z;\hat{Y})$, we show that $\forall Q_1, Q_2 \in \Delta_p$ and $\forall \lambda \in [0,1]$, the following inequality holds: $\mathrm{I}_{Q_\lambda}(Z;\hat{Y})) \leq \lambda \mathrm{I}_{Q_1}(Z;\hat{Y}) + (1-\lambda)\mathrm{I}_{Q_2}(Z;\hat{Y})$.

Note that the marginals are convex in the joint distribution, i.e., $Q_\lambda(z,\hat{y}) = \sum_{s,y} Q_\lambda(z,s,y,\hat{y})$.

This is not necessarily true for conditionals. However, when *some* marginals are fixed, convexity holds for *some* conditionals, i.e., $Q_\lambda(\hat{y}|z) = \sum_{s,y} Q_\lambda(z,s,y,\hat{y})/p(z)$.

Note that the conditional $Q_\lambda(\hat{y}|z)$ is convex in the joint distribution $Q_\lambda(z,s,y,\hat{y})$.

$$
\begin{aligned}
Q_\lambda(\hat{y}|z) &= \sum_{s,y} Q_\lambda(z,s,y,\hat{y})/p(z) = \sum_{s,y} Q_\lambda(\hat{y}|z,s,y)p(s,y,z)/p(z) \\
&= \sum_{s,y} Q_\lambda(\hat{y}|z,s,y)p(s,y|z).
\end{aligned}
$$

Hence, we can show convexity of $\mathrm{I}_Q(Z;\hat{Y})$ in $Q_\lambda(\hat{y}|z)$:

$$
\begin{aligned}
\mathrm{I}_{Q_\lambda}(Z;\hat{Y}) &= \sum_{z,\hat{y}} Q_\lambda(z,\hat{y}) \log\left(\frac{Q_\lambda(z,\hat{y})}{Q_\lambda(z)Q_\lambda(\hat{y})}\right) \\
&= \sum_{z,\hat{y}} Q_\lambda(z)Q_\lambda(\hat{y}|z) \log\left(\frac{Q_\lambda(\hat{y}|z)}{Q_\lambda(\hat{y})}\right) \\
&\overset{(a)}{=} \sum_{z,\hat{y}} p(z)(\lambda Q_1(\hat{y}|z) + (1-\lambda)Q_2(\hat{y}|z)) \log\left(\frac{\lambda Q_1(\hat{y}|z) + (1-\lambda)Q_2(\hat{y}|z)}{\lambda Q_1(\hat{y}) + (1-\lambda)Q_2(\hat{y})}\right) \\
&\overset{(b)}{\leq} \lambda \sum_{z,\hat{y}} p(z)Q_1(\hat{y}|z) \log\left(\frac{Q_1(\hat{y}|z)}{Q_1(\hat{y})}\right) + (1-\lambda) \sum_{z,\hat{y}} p(z)Q_2(\hat{y}|z) \log\left(\frac{Q_2(\hat{y}|z)}{Q_2(\hat{y})}\right) \\
&= \lambda \mathrm{I}_{Q_1}(Z;\hat{Y}) + (1-\lambda)\mathrm{I}_{Q_2}(Z;\hat{Y}).
\end{aligned}
$$

Here (a) holds from expressing the linear combinations. Also note that, $Q_\lambda(\hat{y}) = \sum_z Q_\lambda(\hat{y}|z)p(z)$, which can also be expressed as a linear combination. The inequality (b) holds from the log-sum inequality (see Lemma 4).

To prove the convexity of $\mathrm{I}_Q(Z;\hat{Y}|S)$, we show that $\forall Q_1, Q_2 \in \Delta_p$ and $\forall \lambda \in [0,1]$, the following inequality holds: $\mathrm{I}_{Q_\lambda}(Z;\hat{Y}|S) \leq \lambda \mathrm{I}_{Q_1}(Z;\hat{Y}|S) + (1-\lambda)\mathrm{I}_{Q_2}(Z;\hat{Y}|S)$.

Note that the conditional $Q_\lambda(\hat{y}|z,s)$ is convex in the joint distribution $Q_\lambda(z,s,y,\hat{y})$:

$$
\begin{aligned}
Q_\lambda(\hat{y}|z,s) &= \sum_y Q_\lambda(\hat{y},z,s,y)/p(z,s) = \sum_y Q_\lambda(\hat{y}|z,s,y)p(y,z,s)/p(z,s) \\
&= \sum_y Q_\lambda(\hat{y}|z,s,y)p(y|z,s).
\end{aligned}
$$

Hence, we can show convexity of $I_Q(Z; \hat{Y}|S)$ in $Q_\lambda(\hat{y}|z,s)$:

$$
\begin{aligned}
I_{Q_\lambda}(Z; \hat{Y}|S) &= \sum_{z,s,\hat{y}} Q_\lambda(z,s,\hat{y}) \log \left( \frac{Q_\lambda(\hat{y}|z,s)}{Q_\lambda(\hat{y}|s)} \right) \\
&\stackrel{(a)}{=} \sum_{z,s,\hat{y}} p(z,s) \left( \lambda Q_1(\hat{y}|z,s) + (1-\lambda)Q_2(\hat{y}|z,s) \right) \log \left( \frac{\lambda Q_1(\hat{y}|z,s) + (1-\lambda)Q_2(\hat{y}|z,s)}{\lambda Q_1(\hat{y}|s) + (1-\lambda)Q_2(\hat{y}|s)} \right) \\
&\stackrel{(b)}{\leq} \lambda \sum_{z,s,\hat{y}} p(z,s) Q_1(\hat{y}|z,s) \log \left( \frac{Q_1(\hat{y}|z,s)}{Q_1(\hat{y}|s)} \right) + (1-\lambda) \sum_{z,s,\hat{y}} p(z,s) Q_2(\hat{y}|z,s) \log \left( \frac{Q_2(\hat{y}|z,s)}{Q_2(\hat{y}|s)} \right) \\
&= \lambda I_{Q_1}(Z; \hat{Y}|S) + (1-\lambda) I_{Q_2}(Z; \hat{Y}|S).
\end{aligned}
$$

The equality $(a)$ holds from linear combinations of $Q_\lambda(\hat{y}|s) = \sum_z Q_\lambda(\hat{y}|z,s)p(z|s)$. The inequality $(b)$ holds due to the application of the log-sum inequality (see Lemma 4). $\qquad \square$

## F  EXPANDED EXPERIMENTAL SECTION

This section includes additional results, expanded tables, figures, and details that provide a more comprehensive understanding of our study.

**Dataset.** We consider the following datasets:

*(1) Synthetic dataset:* A 2-D feature vector $X = (X_0, X_1)$ follows a distribution given by $X|_{Y=1} \sim \mathcal{N}((2,2), \left[ \begin{smallmatrix} 5 & 1 \\ 1 & 5 \end{smallmatrix} \right])$, $X|_{Y=0} \sim \mathcal{N}((-2,-2), \left[ \begin{smallmatrix} 10 & 1 \\ 1 & 3 \end{smallmatrix} \right])$. Assume $Z$ is a binary sensitive attribute such that $Z = 1$ if $X_0 > 0$, else $Z = 0$, to encode dependence of $X_0$ with $Z$.

*(2) Adult dataset:* The Adult dataset is a publicly available dataset in the UCI repository based on 1994 U.S. census data (Dua & Graff, 2017). The goal is to predict whether an individual earns more or less than \$50,000 per year based on features such as occupation, marital status, and education. We select *gender* as a sensitive attribute, with men as $Z=1$ and women as $Z=0$.

**Client Distribution.** We strategically partition our datasets across clients to examine scenarios characterized by Unique, Redundant, and Masked Disparities.

*Scenario 1: Uniform Distribution of Sensitive Attributes Across Clients.* The sensitive attribute $Z$ is independently distributed across clients, i.e., $Z \perp\!\!\!\perp S$. We randomly distribute the data across clients.

*Scenario 2: High Heterogeneity in Sensitive Attributes Across Clients.* We split to observe heterogeneity in the distribution of sensitive attributes across clients, i.e., $Z = S$ with a probability $\alpha$. For instance, when $\alpha = 0.9$, the client with $S = 0$ consists of 90% women, while the client with $S = 1$ is composed of 90% men. For the Adult dataset, we use $\alpha = \Pr(Z = 0|S = 0)$ as a parameter to regulate this heterogeneity.

*Scenario 3: High Synergy Level Across Clients.* The true label $Y \approx Z \oplus S$. To emulate this scenario, we partition the data such that client $S = 0$ possesses data of males ($Z = 1$) with true labels $Y = 1$ and females ($Z = 0$) with true labels $Y = 0$. Conversely, client $S = 1$ contains the remaining data, i.e., males with $Y = 0$ and females with $Y = 1$.

We introduce the *synergy level* (Definition 10) to measure alignment to $Y = Z \oplus S$.

**Definition 10** (Synergy Level ($\lambda$)). *The synergy level $\lambda \in [0,1]$ of a given dataset and client distribution is defined as the probability that the true label $Y$ is aligned with $Z \oplus S$,*

$$ \lambda = \Pr(Y = Z \oplus S), $$

*where $\lambda = 1$ implies perfect alignment between $Y$ and $Z \oplus S$, and $\lambda = 0$ implies zero alignment.*

To set $\lambda$ when splitting data across clients, we first split with perfect XOR alignment and then shuffle fractions of the dataset between clients.

*Adult Heterogeneous Split.* In Fig. 3 *(fourth row)*, we split the Adult dataset to capture various disparities simultaneously. We set the synergy level $\lambda = 0.8$ (see Definition 10). Due to the nature of the Adult dataset, this introduces some correlation between the sensitive attribute $Z$ and client $S$.

### F.1 EXPERIMENT A: ACCURACY-GLOBAL-LOCAL-FAIRNESS TRADE-OFF PARETO FRONT.

To study the trade-offs between model accuracy and different fairness constraints, we plot the Pareto frontiers for the AGLFOP. We solve for maximum accuracy $(1 - err)$ while varying global and local fairness relaxations $(\epsilon_g, \epsilon_l)$. We present results for synthetic and Adult datasets as well as PID terms for various data splitting scenarios across clients. The three-way trade-off among accuracy, global, and local fairness can be visualized as a contour plot (see Fig. 3).

For the Adult dataset, we restrict our optimization space $\Delta_p$ to lie within the convex hull derived by the False Positive Rate (FPR) and True Positive Rate (TPR) of an initially trained classifier (trained using FedAvg). This characterizes the accuracy-fairness for all derived classifiers from the original trained classifier (motivated by the post-processing technique from Hardt et al. (2016)). The convex hall characterizes the distributions that can be achieved with any derived classifier. The convex hull for each protected group is composed of points, including $(0,0)$, $(TPR, FPR)$, $(\overline{TPR}, \overline{FPR})$ and $(1,1)$, where $\overline{TPR}$ and $\overline{FPR}$ denote the true positive and false positive rates of a predictor that inverts all predictions for a protected group. Future work could explore alternative constraints for various specialized applications.

### F.2 EXPERIMENT B: DEMONSTRATING DISPARITIES IN FEDERATED LEARNING SETTINGS.

In this experiment, we investigate the PID of disparities in the Adult dataset trained within a FL framework. We employ the *FedAvg* algorithm (McMahan et al., 2017) for training.

**Setup.** Our FL model employs a two-layer architecture with 32 hidden units per layer, using ReLU activation, binary cross-entropy loss, and the Adam optimizer. The server initializes the model weights and distributes them to clients, who train locally on partitioned datasets for 2 epochs with a batch size of 64. Client-trained weights are aggregated server-side via the FedAvg algorithm, and this process iterates until convergence. Evaluation metrics are estimated using the `dit` package (James et al., 2018), which includes PID functions for decomposing Global and Local Disparities into Unique, Redundant, and Masked Disparity, following the definition from Bertschinger et al. (2014).

We analyze the following scenarios:

**PID of Disparity Across Various Splitting Scenarios.** We partition the dataset across two clients, each time varying the level of sensitive attribute heterogeneity ($\alpha = \Pr(Z = 0 | S = 0)$). The Adult dataset exhibits a gender imbalance with a male-to-female ratio of $0.33 : 0.67$. Consequently, $\Pr(Z = 0) = 0.33$, making $\alpha = \Pr(Z = 0 | S = 0) = 0.33$ the independently distributed case.

In this first setup, we distribute the *sensitive attribute uniformly across clients (splitting scenario 1)* and employ FedAvg for training. The FL model achieves an accuracy of $84.45\%$ with a Global Disparity of $0.0359$ bits and a Local Disparity of $0.0359$ bits. The PID reveals that the Unique Disparity is $0.0359$ bits, with both Redundant and Masked Disparities being negligible. This aligns with our centralized baseline, indicating that the disparity originates exclusively from the dependency between the model's predictions and the sensitive attributes, rather than being influenced by $S$.

When the dataset is split to introduce *high heterogeneity in sensitive attributes across clients (splitting scenario 2)*, the resulting FL model exhibits a Global Disparity of $0.0431$ bits and a Local Disparity of $0.0014$ bits. PID reveals a Redundant Disparity of $0.0431$ bits and a Masked Disparity of $0.0014$ bits, with no Unique Disparity.

Next we split and train according to *splitting scenario 3 ($\lambda = 0.9$)*. The trained model reports a Local Disparity of $0.1761$ bits and a Global Disparity of $0.0317$ bits. The PID decomposition shows a Masked Disparity of $0.1761$ bits and a Redundant Disparity of $0.0317$ bits, with no Unique Disparity observed. The emergence of non-zero Redundant Disparity is attributable to the data splitting, which consequently leads to $I(Z; S) = 0.2409$ bits.

We summarize the three scenarios in Fig. 4. Additionally, we evaluate the effects of using a naive local disparity mitigation technique on the various disparities present.

**Effects of Naive Local Fairness Mitigation Technique.** We evaluate the effects of using a naive local disparity mitigation technique on the various disparities present. This is achieved by incorporating a statistical parity regularizer to the loss function at each individual client:

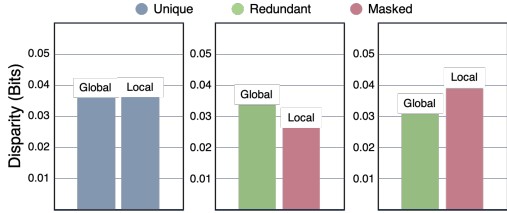

Figure 5: Plot demonstrating scenarios with Unique, Redundant, and Masked Disparities for the Adult dataset 5 client case. Difficulty in splitting to achieve pure Redundant and Masked Disparity due to the proportion of labels in the dataset.

```
client_loss = client_cross_entropy_loss + β client_fairness_loss.
```

We use an implementation from FairTorch package (Akihiko Fukuchi, 2021). The term $\beta$ is a hyperparameter that trades off between accuracy and fairness. We use $\beta = 0.1$ to maintain similarly accurate models. The results are presented in Table 1.

Table 1: Table illustrates the effects of using a naive local disparity mitigation technique on the various scenarios. It proved efficacious only when Unique Disparity is present (*scenario 1*). However, with high redundancy or synergy (*scenarios 2 &3*), the utilization of the disparity mitigation technique exacerbated disparities.

|            | Loc.   | Glob.  | Uniq.  | Red.   | Mas.   |
|------------|--------|--------|--------|--------|--------|
| Scenario 1 | 0.0359 | 0.0359 | 0.0359 | 0.0000 | 0.0000 |
| + fairness | 0.0062 | 0.0062 | 0.0062 | 0.0000 | 0.0000 |
| Scenario 2 | 0.0014 | 0.0431 | 0.0000 | 0.0431 | 0.0014 |
| + fairness | 0.0110 | 0.0626 | 0.0000 | 0.0626 | 0.0110 |
| Scenario 3 | 0.1761 | 0.0317 | 0.0000 | 0.0317 | 0.1761 |
| + fairness | 0.0935 | 0.0418 | 0.0053 | 0.0365 | 0.0882 |

**PID of Disparity under Heterogeneous Sensitive Attribute Distribution.** We analyze the PID of Local and Global Disparities under different sensitive attribute distributions across clients. We train the model with two clients, each having equal-sized datasets. We use $\alpha = \Pr(Z = 0 | S = 0)$ to represent sensitive attribute heterogeneity. Note that for a fixed $\alpha$, the proportions of sensitive attributes at the other client are fixed. For example since $\Pr(Z = 0) = 0.33$ for the Adult dataset, $\alpha = 0.33$ results in even distribution of sensitive attributes across the two clients. Our results are summarized in Fig. 4 and Table 2. We also provide results for 10 federating clients in Table 4.

Table 2: The PID of Global and Local Disparity for varying sensitive attribute heterogeneity $\alpha$

| $\alpha$ | I(Z;S) | Local  | Global | Unique | Redundant | Masked | I($\hat{Y}$;S) | Accuracy |
|----------|--------|--------|--------|--------|-----------|--------|----------------|----------|
| 0.1      | 0.1877 | 0.0262 | 0.0342 | 0.0000 | 0.0342    | 0.0262 | 0.0080         | 86.54%   |
| 0.2      | 0.0575 | 0.0336 | 0.0364 | 0.0064 | 0.0301    | 0.0273 | 0.0028         | 86.95%   |
| 0.3      | 0.0032 | 0.0363 | 0.0365 | 0.0332 | 0.0032    | 0.0031 | 0.0002         | 86.86%   |
| 0.33     | 0.0000 | 0.0340 | 0.0340 | 0.0340 | 0.0000    | 0.0000 | 0.0000         | 87.34%   |
| 0.4      | 0.0154 | 0.0311 | 0.0319 | 0.0186 | 0.0133    | 0.0125 | 0.0009         | 86.70%   |
| 0.5      | 0.0957 | 0.0368 | 0.0413 | 0.0023 | 0.0390    | 0.0345 | 0.0045         | 86.77%   |
| 0.6      | 0.2613 | 0.0242 | 0.0346 | 0.0000 | 0.0346    | 0.0242 | 0.0104         | 86.61%   |
| 0.66     | 0.4392 | 0.0185 | 0.0325 | 0.0000 | 0.0325    | 0.0185 | 0.0140         | 86.36%   |

**Observing Levels of Masked Disparity.** We aim to gain a deeper understanding of the circumstances with Masked Disparities. Through scenario 3, we showed how high Masked Disparities can occur. However, the level of synergy portrayed in the example may not always be present in practice. We attempt to quantify this using a metric *synergy level*. The synergy level ($\lambda$) measures how closely the true labels $Y$ align with the XOR of $Z$ and $S$ (see Definition 10). To achieve a high synergy level, we

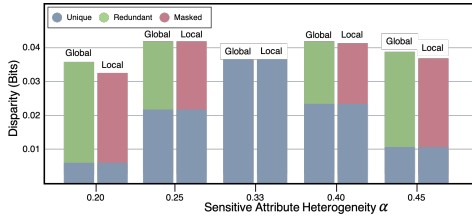

Figure 6: Plot showing the PID of disparities when the data is near i.i.d. among $K = 10$ clients. All types of disparities can be observed. The value $\alpha = 0.33$ represents the case where the data is i.i.d. and only Unique Disparity is observed.

apply the method outlined in Scenario 3. To decrease $\lambda$, we randomly shuffle data points between clients until the synergy level reaches 0. We conduct experiments with varying levels of synergy to observe the impact on the Masked Disparity. The results are summarized in Fig. 4 and Table 3.

Table 3: PID of Global and Local Disparity under varying synergy levels $\lambda$

| $\lambda$ | $I(Z;S)$ | Loc. | Glob. | Uniq. | Red. | Mas. | $I(\hat{Y};S)$ | Acc. | $I(Z;\hat{Y}|S=0)$ | $I(Z;\hat{Y}|S=1)$ |
|---|---|---|---|---|---|---|---|---|---|---|
| 0 | 0.0035 | 0.0402 | 0.0373 | 0.0338 | 0.0035 | 0.0063 | 0.0005 | 85.12% | 0.0196 | 0.0608 |
| 0.25 | 0.0113 | 0.0486 | 0.0419 | 0.0308 | 0.0111 | 0.0178 | 0.0009 | 85.54% | 0.0819 | 0.0152 |
| 0.5 | 0.0299 | 0.0536 | 0.0335 | 0.0127 | 0.0208 | 0.0410 | 0.0033 | 85.24% | 0.1056 | 0.0017 |
| 0.75 | 0.0846 | 0.0932 | 0.0366 | 0.0023 | 0.0343 | 0.0909 | 0.0068 | 85.26% | 0.0024 | 0.1840 |
| 1 | 0.2409 | 0.1644 | 0.0149 | 0.0000 | 0.0150 | 0.1644 | 0.0201 | 84.30% | 0.0839 | 0.2450 |

**Multiple Client Case.** We examine scenarios involving multiple clients. Observations are similar to the two-client case previously studied. To observe a high Unique Disparity, sensitive attributes need to be identically distributed across clients. To observe the Redundant Disparity, there must be some dependency between clients and a specific sensitive attribute, meaning certain demographic groups are known to belong to a specific client. The Masked Disparity can be observed when there is a high level of synergy or XOR behavior between variables $Z$ and $S$. Note that since S is no longer binary, we can convert its decimal value to binary and then take the XOR.

We experiment with $K = 5$ clients and examine the three disparities. To observe the Unique Disparity by randomly distributing the data among clients. For Redundant Disparity, we divide the data such that the first two clients are mostly females and the remaining three clients are mostly males. For Masked Disparity, we distribute the data similarly to scenario 3 (see Fig. 5 and Fig. 6).

**Additional Insights from Experiments.** When data is uniformly distributed across clients, Unique Disparity is dominant and contributes to both global and local unfairness (see Fig. 4 *Scenario 1*: model trained using FedAvg on the adult dataset and distributed uniformly across clients). In the trade-off Pareto Front (see Fig. 3, *row 1*), we see that both local and global fairness constraints have balanced tradeoffs with accuracy. The PID decomposition (Fig. 4, *row 1, column 2,3,4*) explains this as we see the disparity is mainly Unique Disparity, with little Redundant or Masked Disparity. The Unique Disparity highlights where Local and Global Disparity agree.

In the case with sensitive attribute heterogeneity (sensitive attribute imbalance across clients). We observe mainly Redundant Disparity (see Fig. 4, *scenario 2* and *middle*), this is a globally unfair but locally fair model (recall Proposition 1). Observe in the tradeoff plot (see Fig. 3, *row 2*) that the accuracy trade-off is with mainly global fairness (an accurate model could have zero Local Disparity but be globally unfair).

In the cases with sensitive-attribute synergy across clients. For example, in a two-client case (one client is more likely to have qualified women and unqualified men and vice versa at the other client). We observe that the Mask Disparity is dominant (see Fig. 4, *Scenario 3*). The trade AGLFOP tradeoff plot (see Fig. 3, *row 3*) is characterized by Masked Disparity with trade-offs mainly between local fairness and accuracy (an accurate model could have zero Global Disparity but be locally unfair). The Redundant and Masked Disparity highlights where Local and Global Disparity disagree.

Table 4: PID of Global & Local Disparity for various sensitive attribute distributions across 10 clients.

| $\alpha$ | Unique | Redundant | Masked | Global | Local | Accuracy |
|---|---|---|---|---|---|---|
| 0.25 | 0.0219 | 0.0190 | 0.0178 | 0.0409 | 0.0409 | 84.85% |
| 0.33 | 0.0376 | 0.0000 | 0.0000 | 0.0376 | 0.0376 | 85.58% |
| 0.4 | 0.0268 | 0.0141 | 0.0137 | 0.0410 | 0.0405 | 84.85% |
| 0.45 | 0.0107 | 0.0289 | 0.0270 | 0.0390 | 0.0377 | 84.85% |

The AGLFOP provides the theoretical boundaries trade-offs, capturing the optimal performance any model or FL technique can achieve for a specified dataset and client distribution. For example, say one wants a perfectly globally fair and locally fair model, i.e., ($\epsilon_g = 0$, $\epsilon_l = 0$). Under high sensitive attribute heterogeneity (see Fig. 3, *row 2*), they cannot have a model that does better than 64%.

