# OpenReview forum: "Demystifying Local & Global Fairness Trade-offs in Federated Learning Using Partial Information Decomposition"
_ICLR.cc/2024/Conference — ICLR 2024 poster_

### Official Review · Reviewer_GNty · 2023-10-26

**Soundness:** 3 good
**Presentation:** 2 fair
**Contribution:** 4 excellent
**Rating:** 8
**Confidence:** 3

**Summary:**

The paper considers the problem of trying to achieve fairness in a federated learning setting. There are multiple data sets that are all privately held and a model is trained for each one. The questions are: When are the models fair on each data set? And, when are the models fair on all the data?

The paper relates fairness in both settings and mutual information. In particular, they show that mutual information between the  model's prediction and the sensitive attribute is an upper bound on the square of statistical parity (Lemma 1). They also define "local disparity" as mutual information conditioned on the particular machine. They then analyze mutual information and show it comes from three sources: information only in the predictions or sensitive attributes, information in both individually, or information in both together.

They prove necessary/sufficient conditions about when mutual information is low depending on the sources of mutual information. I did not check the appendix for the proofs of these results.

They introduce a convex optimization problem for minimizing classification error subject to constraints that the mutual information and local information are low. They they solve the problem experimentally for different datasets and visualize the results.

**Strengths:**

• The paper investigates the problem of relating global and local fairness in the federated learning setting. According to them (I have not checked), the problem has not been studied before.

• The use existing literature on partial information decomposition to identify sources of mutual information.

• There are a ton of lemmas and theorems about when mutual information is small. It appears very comprehensive but I'd like a more direct narrative about what I should be surprised and impressed with.

• I like the idea of the convex optimization problem and optimizing for error under mutual information constraints.

• The experiments seem very comprehensive in terms of data sets and different settings of data distributions across clients.

**Weaknesses:**

• They don't persuade me that mutual information is the "right" notion of fairness. Lemma 1 establishes that mutual information is an *upper bound* on statistical parity but it could be a loose upper bound.

• I think the presentation is difficult to follow and the paper should be rewritten in the following ways:
- Give an example of the way the theorems and lemmas are proved.
- I was confused by the general approach until I read the examples in Section 3.1. These examples aren't results so I think they should be moved up to the preliminaries section to facilitate understanding.
- Lemma 1 and Lemma 2 are results proved by the authors but they appear in the preliminaries section. This was confusing to me especially because there was no discussion of how they were proved.
- Unique information is used in the preliminaries before it is defined in Definition 3. I didn't find this definition helpful and I don't see similar definitions for redundant information or synergistic information. It would be great if you could define these three quantities in a direct and similar way. I don't know if this is true but maybe something like I(Z,Y|A \cap B) is redundant information and I(Z,Y |A \cup B) is synergistic information. I found the notation you used excessive.
- I'm not sure from reading the main result section if the proofs of the theorems/lemmas following trivially from the definitions or not. Please make this clear with a proof of one of them.

• The convex optimization section is very short. I think you should restructure to spend more space here given that it's one of your contributions.

• I got the sense that the experiments were comprehensive but I was missing a discussion about what was interesting here. It'd be great to highlight interesting observations and findings from the experiments. Examples of how the mutual information perspective gives insight into local and global fairness would be great.

**Questions:**

Lemma 1 upper bounds statistical parity with mutual information, how loose is the upper bound?

What are interesting observations from your expeirments?

---

> ### Author Response · Authors · 2023-11-17
> **Response to Reviewer GNty**
>
> Thank you for your valuable feedback. We have incorporated the suggested changes into our paper to improve the clarity and would love to hear your feedback!
>
> ---
>
> **Presentation of the paper**
>
> We have rearranged the paper as follows:
>
> Preliminary
>
> * Background on Federated learning
>
> * Background on Partial Information Decomposition
>
> * Intuitive example to understand PID terms
> * Formal PID definitions
>
> Main Results
>
> * Formalizing Global and Local Disparity in FL, Lemma 1, and 2
>
> * PID of Global and Local Disparity Proposition 1
>
> * Canonical examples to understand decomposition
>
> * Fundamental limits on tradeoffs between Local and Global Disparity
>
> * An optimization framework for exploring the accuracy-fairness trade-off
>
> We have provided some intuitive examples in the preliminary section to facilitate understanding of PID. We have briefly described a few proofs in the main paper and referred to the location of full proofs in the Appendix. Section 3.2 on fundamental limits has been updated to highlight key results (Theorem 3 and previous Lemma 3 merged, Interaction information definition has been moved to Appendix B). We have further elaborated the section on convex optimization formulation (AFGLOP).
>
> ---
>
> **Mutual information as a fundamental metric for assessing fairness**
>
> Mutual information, denoted by $I(A;B)$, quantifies the dependency between two random variables, A and B (it captures correlation as well as all non-linear dependencies). Within the scope of fairness, consider the sensitive attribute $ Z $ (such as gender or race) and the model's predictions $ \hat{Y} $. The mutual information between $ Z $ and $ \hat{Y} $ provides a measure of their dependence. $I(Z; \hat{Y}) = 0 $ if and only if $ Z $ and $ \hat{Y} $  independent.  Conversely, a high mutual information value indicates a strong correlation between the model's predictions and the sensitive attribute, signaling model unfairness. Mutual information has been explored in fairness in the context of centralized machine learning in [1,2,3].
>
> A recent work [4] provides another interesting interpretation of  $I(Z,\hat{Y})$ in fairness as the accuracy of predicting $ Z $ from $ \hat{Y} $ (or the expected probability of error in correctly guessing $Z$ from $\hat{Y}$). Even in information bottleneck literature [5], mutual information has been interpreted as a measure of how well one random variable predicts (or, aligns with) the other.
>
> While other works [3] have used mutual information measures to enforce statistical parity, we are the first to bring Prinsker’s Inequality [4] in the context of fairness to show that the statistical parity gap is actually upper bounded by the square root of mutual information (see Lemma 1).
>
> **Regarding the tightness of this Lemma 1**: The only inequality used in the proof of Lemma 1 is the Pinsker’s inequality:  $TV(P,Q) \leq \sqrt{0.5KL(P,Q)}$ [6]. Pinsker’s inequality is known to be tight for $KL\leq 1$  (we refer to a short paper on the Tightness of Pinsker’s Bound [6]). We have added Remark 5 in Appendix B Page 15 to highlight this.
>
> *Remark on Tightness of Lemma 1:*
> Since our proof exclusively utilizes Pinsker's inequality, their tightness is equivalent. The mutual information (which can also be defined as a KL divergence) is upper bounded by the entropy $H(\hat{Y})$, that is $ I(Z;\hat{Y}) \leq \text{min}$ { $H(Z), H(\hat{Y})$}$ \leq H(\hat{Y})$. In binary classification  $ H(\hat{Y}) \leq 1 $. Hence, we have $ I(Z;\hat{Y}) \leq 1 $  which is aligned with the known tight regime of Pinsker's inequality (i.e., $KL \leq 1$) [6]. The inequality gets tighter with smaller mutual information $I(Z;\hat{Y})$ values [6].
>
> ---
>
> **Definitions for redundant information or synergistic information**
>
> We have made some edits in the preliminary section on Page 3 to clarify this further.
>
> We only provide the formal definition of $\text{Uni(Z:A|B)}$ because defining any one of the PID terms suffices to get the others. $\text{Red}(Z:A, B)$ is the sub-volume between $I(Z;A)$ and $I(Z;B)$ (see Fig.1). Hence,  $\text{Red}(Z:\hat{Y}, S) = I(Z;\hat{Y}) - \text{Uni}(Z: \hat{Y}| S)$ and $\text{Syn}(Z:\hat{Y},S) =  I(Z; \hat{Y}, S) - \text{Uni}(Z: \hat{Y}|S) -\text{Uni}(Z: \hat{Y}|S)-\text{Red}(Z:\hat{Y}, S)$ (from Equation 1).
>
> ---
>
> [1]Fairness-Aware Classifier with Prejudice Remover Regularizer https://link.springer.com/chapter/10.1007/978-3-642-33486-3_3
>
> [2] A Fair Classifier Using Mutual Information, https://ieeexplore.ieee.org/stamp/stamp.jsp?tp=&arnumber=9174293
>
> [3] INFOFAIR: Information-Theoretic Intersectional Fairness  https://arxiv.org/pdf/2105.11069.pdf
>
> [4] Can Information Flows Suggest Targets for Interventions in Neural Circuits? https://arxiv.org/abs/2111.05299
>
> [5]The Information Bottleneck Problem and Its Applications in Machine Learning  https://arxiv.org/pdf/2004.14941.pdf
>
> [6] https://blog.wouterkoolen.info/Pinsker/post.pdf
>
> **Continues 1/3**

---

> ### Author Response · Authors · 2023-11-17
> **Response to Reviewer GNty**
>
> **How Theorems and Lemmas are proved**
>
> We have now briefly described a few proofs in the main paper and referred to the location of full proofs in the appendix.
>
> The proof of Lemma 1 is non-trivial as it requires Pinsker’s inequality.  Since Mutual information can be written in KL divergence, i.e., $I(Z; \hat{Y}) =  KL(P(\hat{Y}, Z)|| P(\hat{Y})P(Z))$. We show that $TV(P(\hat{Y}, Z), P(\hat{Y})P(Z)) = 2 \alpha (1-\alpha)|SP|$, where $\alpha = P(Z=0)$ (Complete proof in Appendix B).
>
> The proof of Lemma 2 leverages writing the definition of conditional mutual information in terms of the joint and marginal probability distributions (Complete proof in Appendix B).
>
> For the proof of Proposition 1, Global disparity decomposition (2) follows from the relationship between different PID terms, Local disparity decomposition (3) requires chain rule of mutual information. Showing non-negativity is challenging with some additional steps needed. Theorem 1 and 2  follow from Proposition 1 but particularly require non-negativity with additional steps.
>
> As an example, in Theorem 1:
>     As local disparity $I(Z;\hat{Y}|S) \rightarrow 0$, then $\text{Uni}(Z:\hat{Y}|S) \rightarrow 0$ and $\text{Syn}(Z:\hat{Y},S) \rightarrow 0$, therefore the global disparity $I(Z;\hat{Y}) \rightarrow \text{Red}(Z:\hat{Y},S) \geq 0$.
>
> Theorem 3 also follows from Proposition 1. However, the sufficiency step requires us to leverage PID of $I(Z;S)$ and the mutual information independence property.
>
> In Theorem 4, using Proposition 1, the argument goes as follows.
>
> By leveraging the PID of $I(Z;S|\hat{Y}) = \text{Uni}(Z:S|\hat{Y})+\text{Syn}(Z:\hat{Y},S)$
>
> Markov chain $Z - \hat{Y} - S$ implies, $I(Z;S|\hat{Y}) = 0 $. Hence, $ \text{Syn}(Z:\hat{Y},S)=0$.
>
> Rest of proof follows from nonnegative property of PID terms: $I(Z;\hat{Y}|S)=\text{Uni}(Z:\hat{Y}|S)  \leq \text{Uni}(Z:\hat{Y}|S)  + \text{Red}(Z:\hat{Y},S)  = I(Z;\hat{Y}).$
>
> In the proof of Theorem 6, we show the AFGLOP is a convex optimization problem by showing the objective and mutual information constraints are convex functions on a convex set $\Delta_p$. This required the log sum inequality and an observation that some conditional distributions are convex in the joint distribution $(Z,S,Y,\hat{Y})$ (Complete proof in Appendix E).
>
> **Continues 2/3**

---

> ### Author Response · Authors · 2023-11-17
> **Response to Reviewer GNty**
>
> **On the interesting observations from our experiments**
>
> We are grateful for the reviewer's recognition of the thoroughness of our experiments.  Our experimental results reinforces our findings and highlight the various scenarios in which unique, redundant, and masked disparities can occur in a practical setting. Our information-theoretic framework provides a nuanced understanding of the three sources of disparity in FL. Our experiments offer insights into the agreement and disagreement between local and global fairness under various data distributions. Our experiments and theoretical results show that depending on the data distribution achieving one (global or local fairness) can often come at the cost of the other (disagreement), emphasizing the need for global and local fairness to be considered together when addressing group fairness problems in FL. The way data is distributed across clients significantly impacts the type of disparity (unique, redundant, masked) that dominates. Our optimization framework establishes the accuracy-fairness tradeoffs for a given dataset and client distribution. We have updated the discussions on Page 9.
>
> ---
>  We summarize more observations here (also added to Appendix F on Page 25):
>
> When data is uniformly distributed across clients, unique disparity is dominant and contributes to both global and local unfairness (see Figure 4 Scenario 1: model trained using FedAvg on the adult dataset and distributed uniformly across clients). In the trade-off Pareto Front  (Figure 3, Row 1), we see that both local and global fairness constraints have balanced tradeoffs with accuracy. The PID decomposition (Figure 4, Row 1, column 2,3,4) explains this as we see the disparity is mainly unique disparity, with little redundant or masked disparity. The unique disparity highlights where local and global disparity are in agreement.
>
> In the case with sensitive attribute heterogeneity (sensitive attribute imbalance across clients). The disparity observed is mainly redundant disparity (see Figure 4 scenario 2 and middle), this is a globally unfair but locally fair model (recall proposition 1). Observe in the tradeoff plot (see Figure 3, Row 2) that the accuracy trade-off is with mainly global fairness (an accurate model could have zero local disparity but be globally unfair).
>
> In the cases with sensitive-attribute synergy across clients. For example, in a two-client case (one client is more likely to have qualified women and unqualified men and vice versa at the other client). We observe that the mask disparity is dominant (see Figure 4 Scenario 3). The trade AGLFOP tradeoff plot (see Figure 3, Row 3) is characterized by Masked Disparity with trade-offs mainly between local fairness and accuracy (an accurate model could have zero global disparity but be locally unfair).
>
> The AGLFOP provides the theoretical boundaries trade-offs, capturing the optimal performance any model or FL technique can achieve for a specified dataset and client distribution. For example, say one wants a perfectly globally fair and locally fair model, i.e., ($\epsilon_g=0$, $\epsilon_l=0$). Under high sensitive attribute heterogeneity (see Figure 3 Row 2), they cannot have a model that does better than 64% accuracy on the dataset.
>
> **3/3**

---

> > ### Comment · Reviewer_GNty · 2023-11-22
> >
> > Thank you for your *very* thorough responses.
> >
> > I now see how your upper bound is quite tight (depending on mutual information).
> >
> > I appreciate your engagement with my restructuring suggestions.
> >
> > I understand that your experiments broadly give insight into the trade off between global and local fairness. I would love to see *specific* examples. Such as: "In X data set, we see this trade off between synergistic and redundant information indicating Y concept which intuitively makes sense for Z reason."
> >
> > Based on your thorough responses and with the understanding that you'll add more specific examples such as the one above, I'll increase my rating to accept.

---

> > > ### Author Response · Authors · 2023-11-23
> > >
> > > We thank you for your valuable feedback and increasing your rating. We will incorporate specific examples as suggested to further highlight the insight of our tradeoff analysis.

---

### Official Review · Reviewer_rTkq · 2023-10-31

**Soundness:** 3 good
**Presentation:** 3 good
**Contribution:** 3 good
**Rating:** 6
**Confidence:** 2

**Summary:**

This work presents an information-theoretic perspective on group fairness trade-offs in federated learning (FL) with respect to sensitive attributes. This paper leverages partial information decomposition to identify three sources of unfairness in FL. They introduce AGLFOP, a convex optimization that defines the theoretical limits of accuracy and fairness trade-offs, identifying the best possible performance any FL strategy can attain given a dataset and client distribution.

**Strengths:**

1. This paper studies group fairness from an information theory perspective, which is valuable for the community to understand the group fairness of FL.
2. The decomposition result is interesting.

**Weaknesses:**

1. Although this paper proposes an optimization framework with Definition 5, it does not give any solution or algorithm for solving the problem.
2. The experiments are somehow weak, both the baseline and dataset are rare. There are other works focusing on FL group fairness like [1,2], and also about fairness and accuracy tradeoffs, like [3], that should be compared.
3. The visualization (table or figure) of accuracy and global-local fairness trade-off results is relatively insufficient relying solely on the Pareto Frontiers shown in Figure 3.
4. It seems of vital importance to properly set the hyper-parameters $\epsilon_g$ and $\epsilon_L$ for the optimal trade-off. The acc-fairness Trade-off figure displayed in Figure 3, lacks discussion based on more experimental settings and datasets.
5. The experiment setting details are not clear, for example, what is the used model and parameter settings?

[1]Ezzeldin Y H, Yan S, He C, et al. Fairfed: Enabling group fairness in federated learning[C]//Proceedings of the AAAI Conference on Artificial Intelligence. 2023, 37(6): 7494-7502.
[2]Papadaki A, Martinez N, Bertran M, et al. Minimax demographic group fairness in federated learning[C]//Proceedings of the 2022 ACM Conference on Fairness, Accountability, and Transparency. 2022: 142-159.
[3]Wang L, Wang Z, Tang X. FedEBA+: Towards Fair and Effective Federated Learning via Entropy-Based Model[J]. ICLR 2023 Workshop ML4IoT.

Minors:
What is the formal definition of global fairness and local fairness?

**Questions:**

1. Could you explain the main difference from [4]?  It seems it is a trivial improvement (Apply the PID analysis on FL) compared with this paper.
2. Could you provide experiments of different baselines and different datasets, in the FL setting (partial client participation of cross-device FL.)
3. Could you provide more results for trade-offs on accuracy and global-local fairness?
4. Could you provide more details about the selection of hyperparameters to ensure optimal trade-off strategy under different data distributions and datasets?

[4] Dutta S, Hamman F. A Review of Partial Information Decomposition in Algorithmic Fairness and Explainability[J]. Entropy, 2023, 25(5): 795.

---

> ### Author Response · Authors · 2023-11-17
> **Response to Reviewer rTkq**
>
> We thank the reviewer for reviewing our paper!
>
> ---
>
> **Clarification on our contributions**
>
> We want to highlight the main contribution of this paper is to provide a **theoretical**  perspective to global and local fairness trade-offs in FL. We use PID to decompose global & local disparity into three sources of unfairness: Unique Disparity, Redundant Disparity, and Masked Disparity. This decomposition separates out the regions of agreement and disagreement of local and global disparity, demystifying their trade-offs. We also formulate a convex optimization framework for quantifying accuracy-global and local fairness trade-offs identifying the best possible performance any FL strategy can attain given a dataset and client distribution.  Our aim is not to provide a state-of-the-art algorithm but to provide a nuanced understanding of the sources of disparity in FL, which can inform the use of disparity mitigation techniques and their effectiveness when deployed in practice. We provide a comprehensive experimental section with various settings of data distributions across clients to support our claims (also acknowledged by reviewer GNty).
>
> ---
>
> **Regarding the comparison with the works [1,2, 3]**
>
> Our paper's aim diverges from these works. While [1,2] predominantly focuses on mitigating unfairness in FL, our work delves into theoretically exploring the fundamental limitations on the trade-offs between accuracy-global and local fairness. Our work establishes the theoretical limits of what any FL technique can achieve in terms of accuracy and fairness given a dataset and client distribution.
>
> Similarly in [3], the definition and scope of **fairness** differ substantially from our work (we highlight the different notions of fairness in FL in the related works section and have cited [3]). [3] concentrates on **client fairness**, emphasizing model performance is consistent across clients. In contrast, our work is centered around **group fairness**, particularly addressing model discrimination against sensitive attributes (e.g, race, gender).
>
> ---
>
> **Clarification on Accuracy-Global and Local Fairness Tradeoff  (W2, W4, Q3, Q4)**
>
> We introduce the Accuracy and Global-Local Fairness Optimality Problem (AGLFOP) as an optimization to delineate the theoretical boundaries of accuracy and fairness trade-offs, capturing the optimal performance any model or FL technique can achieve for a specified dataset and client distribution.
>
> We aim to find the maximum achievable accuracy, quantified as $1 - c(Q)$, within the context of specific constraints on global and local disparities in fairness ($I(Z;\hat{Y}) \leq \epsilon_g$) for global disparity and \($I(Z;\hat{Y}|S) \leq \epsilon_l$) for local disparity. Here, $\epsilon_g$ and $\epsilon_l$ are fairness relaxation constants and not hyperparameters.
> For example, consider the scenario where both $\epsilon_l = 0$ and $\epsilon_g = 0$. This represents a condition of perfect fairness both locally and globally. In such a case, our objective is to determine the highest level of accuracy that a model can achieve while adhering to these ideal global and local fairness constraints.
>
> ---
>
> **W1: Solution and algorithm for solving optimization**
>
> We show that the AGLFOP is a convex optimization problem and hence can be solved efficiently using standard convex solvers (which are well-established and robust, to find globally optimal solutions efficiently). This optimization problem can be computed in any FL environment. Specifically, their computation necessitates the characterization of the joint distribution $\Pr(Z{=}z,S{=}s,Y{=}y)= \Pr(S{=}s)\Pr(Z{=}z|S{=}s)\Pr(Y{=}y|Z{=}z,S{=}s)$, which can be acquired by aggregating pertinent statistics across all participating clients. For instance, $\Pr(S=s)$ denotes the proportion of data at client $s$, $\Pr(Z=z|S=s)$ signifies the fraction of individuals with sensitive attribute $z$ at client $s$, and $\Pr(Y=y|Z=z,S=s)$ represents the proportion of individuals labeled with true label $y$ among those with sensitive attribute $z$ at client $s$.
>
> ---
>
> [1] Fairfed: Enabling group fairness in federated learning, arxiv.org/abs/2201.08304
>
> [2] Minimax demographic group fairness in federated learning, arxiv.org/abs/2201.08304
>
> [3]  Towards Fair and Effective Federated Learning via Entropy-Based Model, arxiv.org/abs/2301.12407
>
> **Continues 1/2**

---

> > ### Comment · Reviewer_rTkq · 2023-11-21
> > **Thanks for your rebuttal**
> >
> > I would like to thank the authors for their detailed explanations. Now I understand that the solution is based on the fact that the proposed AGLFOP is a convex problem and the roles of $\epsilon_g$ and $\epsilon_L$ are the relaxation constants instead of hyperparameters. It is also nice to see the discussion on $\Delta_p$ involved in AGLFOP has been added in the revised manuscript.
> >
> > As this paper is centered around group fairness, is it possible to show the superiority of AGLFOP compared with some baselines of group fairness in the experiments?

---

> ### Author Response · Authors · 2023-11-17
> **Response to Reviewer rTkq**
>
> **W3, Q3: Visualization of accuracy and global-local fairness trade-off using Pareto Frontiers**
>
> The use of Pareto Frontiers is a well-established method for trade-off analysis in optimization, offering a clear and concise representation of the optimal trade-offs between competing objectives. In Figure 3, we provide the AGLFOP Pareto Frontiers showing maximum accuracy (shown as contour) that can be achieved on a dataset and client distribution for a given global and local fairness various relaxation $\epsilon_g$ ($x$-axis) and $\epsilon_l$ ($y$-axis) respectively. This helps visualize the effects of global and local fairness constraints on the accuracy of the model (i.e., Accuracy-global local fairness tradeoff). This method allows for a comprehensive overview of the optimal solutions, facilitating informed decision-making about the trade-offs that are inherent to the model's performance under fairness constraint.
>
>
> ---
>
> **W5: On experimental setting details; model and parameter settings**
>
>  Due to lack of space, we provided detailed descriptions of our experimental setup, including model specifications and parameter settings, in Appendix F2 of our paper.
>
> ---
>
> **Formal Definition of Global Fairness and Local Fairness**
>
> Global fairness is the overall disparity of the model across all clients while local fairness is the disparity of the model at each client. We introduce formal mathematical measures of global fairness in Definition 1 (Global Disparity) and local fairness in Definition 2 (Local Disparity).
>
> ---
>
> **Q1: Difference between Survey Paper and our Paper**
>
> It is important to note that [1] is a survey paper that surveys other papers that apply partial information decomposition (PID) in algorithmic fairness and explainability and provides a taxonomy. PID is a mathematical tool whose origin goes back to [2] and has been used previously in neuroscience. Our contribution is in leveraging PID specifically for the setup of group fairness in FL and using it to decompose disparities as well as formulate a novel convex optimization problem to characterize tradeoffs.
>
> ---
>
> **Q2: On partial client participation of cross-device FL**
>
> Our approach does not make specific assumptions about the training process of the federated learning model, which allows our results to be applicable to scenarios involving partial client participation in cross-device federated learning methods as well.  This flexibility is a significant aspect of our contribution, as it allows our methods to be adapted and applied in diverse federated learning contexts.
>
> ---
>
> [1] Dutta S, Hamman F. A Review of Partial Information Decomposition in Algorithmic Fairness and Explainability.
>
> [2] Nonnegative Decomposition of Multivariate Information, https://arxiv.org/abs/1004.2515
>
> **2/2**

---

> ### Author Response · Authors · 2023-11-23
>
> We thank the reviewer for appreciating our rebuttal!
>
> *The goal of the AGLOP is to theoretically study the tradeoff between accuracy and global local fairness for a specified dataset and client distribution rather than provide a state-or-the-art algorithm*. To the best of our knowledge, there are no suitable baselines to compare with that quantify such fundamental tradeoffs between local and global fairness. Nonetheless, we ran some methods that attempt to minimize only the global fairness on the Adult dataset. For these methods, we measure the global and local disparities for various data splits and see that the accuracy achieved by the models is lower than the maximum accuracy limit for the given local and global disparity relaxation using our proposed optimization. We can include a remark regarding this observation in the paper as recommended by the reviewers.
>
>
>
>
>
>
>
>
> | Method                | **Local Disparity** | **Global Disparity** | **Accuracy** |**AGLFOP**             |
> |-----------------------------|---------------------|----------------------|------------------|-----------------------|
> | FedFB (scenario 1)             | 0.0249     	   |   0.0241            | 81%                | 86%       	|
> | FedFB (scenario 1)             | 0   		  |   0      	       | 75%         | 82%     		 |
> | FairBatch FL (scenario 1)             | 0.0019              | 0.0019               | 81%            | 83%                  |
> | FedAvg (scenario 2)     | 0.0180              | 0.0335               | 84%               | 86%                |
> | FedAvg  (scenario 3)                 | 0.0148              | 0.0067     | 76%           | 78%      |
>
> Scenario 1:  sensitive-attribute independently distributed across clients
>
> Scenario 2: high sensitive-attribute heterogeneity across clients ($\alpha = 0.67$)
>
> Scenario 3: high sensitive-attribute synergy level across clients ($\lambda = 0.90$)
>
> **Methods**
>
> FedAvg (+Local Fair Training): Communication-Efficient Learning of Deep Networks from Decentralized Data, https://proceedings.mlr.press/v54/mcmahan17a.html
>
> FedFB: Improving Fairness via Federated Learning, https://arxiv.org/abs/2110.15545
>
> FairBatch FL (FairBatch Training + Ensemble): FairBatch: Batch Selection for Model Fairness, https://arxiv.org/abs/2012.01696

---

### Official Review · Reviewer_P5hg · 2023-10-31

**Soundness:** 4 excellent
**Presentation:** 4 excellent
**Contribution:** 4 excellent
**Rating:** 8
**Confidence:** 3

**Summary:**

This paper introduces information theoretic tools to interpret the relationship among multiple group fairness trade-offs in federated learning. It is commonly known that global and local fairnesses both contribute to unfairness in federated learning, but their relationship (e.g. whether one implies the other) is unknown. The authors identify three fundamental sources of unfairness, and utilize them to derive fundamental limits on the trade-offs between global and local unfairnesses.

**Strengths:**

This paper provides a novel "solution" for unfairness in federated learning. Even if this issue has been extensively studied, there has been few work attempting to investigate the fundamental root that causes unfairness, let alone giving a theoretical explanation. This work uses a mathematically rigorous tool to give a promising attempt to explain unfairness. The theoretical justifications are rigorous and insightful.

In particular, Theorems 1, 2 and 3 are both conclusive and powerful, so that one may predict the fairness performances based upon those three sources of unfairness.

**Weaknesses:**

Overall this paper is well-written, but in Section 2, it would be great if the authors could provide some more justifications for Definitions 1 and 2, both mathematically and conceptually, even if the definitions are indeed fairly intuitive. This may be immensely helpful for readers especially those who do not have a strong background in information theory.

**Questions:**

1. This was mentioned in the Weaknesses section, and I would be very interested in seeing more explanations for choosing those definitions.

2. Under a concrete data set, how are Uni(), Red(), and Syn() efficiently computed? I might be wrong, but my first impression is that, since they are relevant with mutual information, such computation may be expensive?

---

> ### Author Response · Authors · 2023-11-17
> **Response to Reviewer P5hg**
>
> We are grateful to the reviewer for appreciating our contributions. We thank the reviewer for their valuable feedback regarding the definitions and computations of our measures. We discuss these below and have updated the manuscript (see Appendix A on Page 14).
>
> ---
>
> **Justifications for Definitions 1 and 2 (mathematically and conceptually)**
>
> Let's consider a case involving three banks training a federated model for credit scoring: Bank 1, predominantly serving men; Bank 2, catering to a diverse demographic; and Bank 3, primarily serving women. Global fairness means that the model doesn’t discriminate against any protected group when evaluated across all client (Bank) data. Meanwhile, local fairness means fairness within each client’s data (fairness at each Bank).
>
> In this work, we use mutual information as a measure of the unfairness or disparity of a model.
> Mutual Information has been interpreted as the dependence between sensitive attribute $Z$  and model prediction $\hat{Y}$ (captures correlation as well as all non-linear dependencies). Mutual information is zero if and only if $Z$ and $\hat{Y}$ are independent. This means that if the model’s predictions are highly correlated with sensitive attributes (like gender or race), that’s a sign of unfairness. Mutual information has been explored in fairness in the context of centralized machine learning in [1,2,3].
>
> A recent work [5] provides another interpretation of mutual information $I(Z; \hat{Y})$ in fairness as the accuracy of predicting $Z$ from $\hat{Y}$  (or the expected probability of error in correctly guessing $Z$ from $\hat{Y}$). Even in information bottleneck literature [6], mutual information has been interpreted as a measure of how well one random variable predicts (or, aligns with) the other.
>
> For local fairness, we are interested in the dependence between model prediction $ \hat{Y} $ and sensitive attributes $Z$ at each and every client, i.e., the dependence between $\hat{Y}$ and $Z$ conditioned on the client $S$. For example, the disparity at client $S=1$ (Bank 1) is $I(Z; \hat{Y}|S=1)$ (the mutual information (dependence) between model prediction and sensitive attribute conditioned on client $S=1$ (considering data at client $S=1$). Our measure for local disparity (Definition 2) is the conditional mutual information (dependence) between $Z$ and $\hat{Y}$ conditioned on $S$, denoted as $I(Z;\hat{Y}|S)$. Local disparity $I(Z;\hat{Y}|S) =\sum_s p(s)I(Z; \hat{Y}|S=s)$, is an average of the disparity at each client weighted by the $p(s)$, the fraction of data at client $S=s$.  The local disparity is only zero if and only if all client has zero disparity in their local dataset.
>
> While other works [3] have used mutual information measures to enforce statistical parity, we are the first to bring Prinsker’s Inequality [4] in the context of fairness to show that the statistical parity gap is actually upper bounded by the square root of mutual information (see Lemma 1). Though this is not the main result of the paper, it further justifies the use of mutual information measures to study global and local fairness in federated learning.
>
> We thank the reviewer for their valuable feedback on making our work accessible to readers without strong backgrounds in information theory. We have included a brief background on concepts and definitions in Appendix A on Page 14.
>
>
>
>
> [1]Fairness-Aware Classifier with Prejudice Remover Regularizer https://link.springer.com/chapter/10.1007/978-3-642-33486-3_3
>
> [2] A Fair Classifier Using Mutual Information, https://ieeexplore.ieee.org/stamp/stamp.jsp?tp=&arnumber=9174293
>
> [3] INFOFAIR: Information-Theoretic Intersectional Fairness  https://arxiv.org/pdf/2105.11069.pdf
>
> [4] https://blog.wouterkoolen.info/Pinsker/post.pdf
>
> [5] Can Information Flows Suggest Targets for Interventions in Neural Circuits? https://arxiv.org/abs/2111.05299
>
> [6]The Information Bottleneck Problem and Its Applications in Machine Learning  https://arxiv.org/pdf/2004.14941.pdf
>
>
> **Continues 1/2**

---

> ### Author Response · Authors · 2023-11-17
> **Response to Reviewer P5hg**
>
> **On the Computation of Uni, Red, Syn**
>
> The computation of Uni, Red, Syn disparity necessitates the characterization of the joint distribution $\Pr(Z{=}z,S{=}s,\hat{Y}{=}\hat{y})=\Pr(S{=}s)\Pr(Z{=}z|S{=}s)\Pr(\hat{Y}{=}\hat{y}|Z{=}z,S{=}s)$, which can be acquired by aggregating pertinent statistics across all participating clients.  For instance, $\Pr(S=s)$ denotes the proportion of data at client $s$, $\Pr(Z=z|S=s)$ signifies the fraction of individuals with sensitive attribute $z$ at client $s$, and $\Pr(\hat{Y}=\hat{y}|Z=z,S=s)$ represents the proportion of individuals with predictived label $\hat{y}$ among those with sensitive attribute $z$ at client $s$.
>
>
> The global and local disparity is first measured empirically using the classical definition of mutual information, i.e., $ I(Z; \hat{Y}) = \sum_{z , \hat{y} } p(z, \hat{y}) \log \frac{p(z, \hat{y})}{p(z)p(\hat{y})}$ and conditional mutual information, i.e.,  ${ I(Z; \hat{Y} | S) = \sum_{s, z , \hat{y}} p(s, z, \hat{y}) \log \frac{p(z, \hat{y} | s)}{p(z | s)p(\hat{y} | s)}}$. For the Unique disparity, we use Definition 3 which solves a convex optimization problem over the joint distribution $(Z,\hat{Y}, S)$ keeping the marginals over $(Z,\hat{Y}) $ and $(Z, S)$ fixed (we use an implementation from the Discrete Information Theory [1] python package) as shown in [2].
>
> Defining any one of the PID terms suffices to get the others. $\text{Red}(Z:A, B)$ is the sub-volume between $I(Z;A)$ and $I(Z;B)$ (see Fig.1). Hence,  $\text{Red}(Z:\hat{Y}, S) = I(Z;\hat{Y}) - \text{Uni}(Z: \hat{Y}| S)$ and $\text{Syn}(Z:\hat{Y},S) =  I(Z; \hat{Y}, S) - \text{Uni}(Z: \hat{Y}|S) -\text{Uni}(Z: S|\hat{Y})-\text{Red}(Z:\hat{Y}, S)$ (from Equation 1).
>
> This estimation largely depends on:
>
> (i) the empirical estimators of the probability distributions; and
>
>  (ii) the efficiency of the convex optimization algorithm used for calculating the unique information.
>
> Convex optimization problems are generally well-studied and have efficient solvers. In our experiments on the Adult dataset with $2$,$5$, and $10$ federated clients, estimating the disparity terms was not computationally expensive.  However, the method's computational cost can increase with the number of clients or sensitive attributes. We have highlighted this limitation in the paper (see Discussion on Page 9). There's also emergent interest in the estimation of these measures using Gaussian relaxations [3] or neural networks [4,5].
>
>
> [1] dit: a Python package for discrete information theory, doi.org/10.21105/joss.00738
>
> [2] Quantifying unique information, arxiv.org/abs/1311.2852
>
> [3] Partial Information Decomposition via Deficiency for Multivariate Gaussians, arxiv.org/abs/2105.00769
>
> [4] MINE: Mutual Information Neural Estimation, arxiv.org/abs/1801.04062
>
> [5] Redundant Information Neural Estimation, www.mdpi.com/1099-4300/23/7/922
>
>
>  **2/2**

---

### Author Response · Authors · 2023-11-17
**Global Response**

We are grateful for the constructive feedback provided by the reviewers.  We appreciate Reviewer P5hg's recognition of the **novelty of our solutions** and their commendation of our theoretical justifications as both **rigorous** and **insightful**. We also thank Reviewer rTkq for finding the information theory perspective **valuable to the community** and partial information decomposition result **interesting**. We are thankful to Reviewer GHty for recognizing the **novelty of our problem** and their positive view on our convex optimization formulation (AGLFOP).  We also appreciate their acknowledgment of the **comprehensiveness of our experiments**.

Based on the reviews, we have made several changes to the paper:

* We have included a background on information theory in the Appendix for readers without a strong background in information theory and included concepts and definitions used in the paper.

* We have added a discussion on the estimation of our PID measures.

* We have cited the paper brought up by the Reviewer rTkq.

* We have incorporated several other changes in organization and presentation as recommended by Reviewer GHty.

The updated parts of our main paper are highlighted in **blue** for ease of review. We now provide detailed responses to each reviewer below.

---

### Meta-Review · Area_Chair_BwzT · 2023-12-09

**Metareview:**

The paper addresses the challenge of achieving fairness in this setting with multiple privately held datasets, training models for each, i.e., the federated learning setting.  It explores when models are fair on individual datasets and when they are fair across all datasets, linking fairness, mutual information, and statistical parity. This work offers an information-theoretic perspective on group fairness trade-offs in federated learning, utilizing partial information decomposition to identify three sources of unfairness. The authors present a convex optimization problem, minimizing classification error while constraining mutual information and local information - this helps establish theoretical limits for accuracy and fairness trade-offs in federated learning. Experimental results are often presented.

The paper presents several concrete technical contributions. Among them, revealing the trade-offs between individual and group fairness, and pinging down the three sources of unfairness are powerful results that can provide insights to the fairness community.

**Justification For Why Not Higher Score:**

The paper is dense and is not super easy to follow at places, limiting its broader impact. The authors are also encouraged to add more discussions to the empirical findings, and better relate them to the theoretical findings.

**Justification For Why Not Lower Score:**

The reviewers are unanimous about the paper's technical quality and contributions. The paper presented convincing theoretical and experimental results that can find interest in the relevant research communities. The results also have the potential to kick off more discussions on achieving fairness in the federated learning setting.

---

### Decision · Program_Chairs · 2024-01-16

Accept (poster)